# FRET biosensor uncovers cAMP nano-domains at β-adrenergic targets that dictate precise tuning of cardiac contractility

Nicoletta C. Surdo[1], Marco Berrera[2], Andreas Koschinski[1], Marcella Brescia[1], Matias R. Machado[3], Carolyn Carr[1], Peter Wright[4], Julia Gorelik[4], Stefano Morotti[5], Eleonora Grandi[5], Donald M. Bers[5], Sergio Pantano[3] & Manuela Zaccolo[1,2]

Compartmentalized cAMP/PKA signalling is now recognized as important for physiology and pathophysiology, yet a detailed understanding of the properties, regulation and function of local cAMP/PKA signals is lacking. Here we present a fluorescence resonance energy transfer (FRET)-based sensor, CUTie, which detects compartmentalized cAMP with unprecedented accuracy. CUTie, targeted to specific multiprotein complexes at discrete plasmalemmal, sarcoplasmic reticular and myofilament sites, reveals differential kinetics and amplitudes of localized cAMP signals. This nanoscopic heterogeneity of cAMP signals is necessary to optimize cardiac contractility upon adrenergic activation. At low adrenergic levels, and those mimicking heart failure, differential local cAMP responses are exacerbated, with near abolition of cAMP signalling at certain locations. This work provides tools and fundamental mechanistic insights into subcellular adrenergic signalling in normal and pathological cardiac function.

[1] Department of Physiology, Anatomy and Genetics, University of Oxford, Oxford OX1 3PT, UK. [2] Molecular Pharmacology Centre, Institute of Neuroscience and Psychology, University of Glasgow, Glasgow G12 8QQ, UK. [3] Group of Biomolecular Simulations, Institute Pasteur de Montevideo, Montevideo CP 11400, Uruguay. [4] National Heart and Lung Institute, Imperial Centre for Translational and Experimental Medicine, Imperial College London, London W12 0NN, UK. [5] Department of Pharmacology, University of California Davis, Davis, California 95616, USA. Correspondence and requests for materials should be addressed to M.Z. (email: manuela.zaccolo@dpag.ox.ac.uk).

Physical and emotional stress, via release of catecholamines, activation of G-protein-coupled receptors (GPCRs) and synthesis of 3′-5′-cyclic adenosine monophosphate (cAMP), promotes a series of events, known as the fight-or-flight response, aimed at avoiding potential harm. In the heart this involves activation of β-adrenergic receptors (β-AR) and consequent increases in heart rate (chronotropy), contractile strength (inotropy) and relaxation rate (lusitropy), so that cardiac output can be enhanced dramatically to support the increased oxygen demands of the body. The main effector of cAMP, protein kinase A (PKA), phosphorylates key targets in cardiac myocytes, including L-type $Ca^{2+}$ channels (LTCC) at the plasmalemma, phospholamban (PLB) on the sarcoplasmic reticulum (SR), and troponin I (TPNI) and myosin binding protein C (MyBPC) on the myofilaments. These proteins participate in the coupling of cell excitation to myocyte contraction, a process that is graded by the concentration of intracellular $[Ca^{2+}]$ ($[Ca^{2+}]_i$) at each beat and by adjusting the myofilament $Ca^{2+}$ sensitivity[1].

In addition to the β-AR fight-or-flight response, cAMP also mediates signalling by numerous hormones, neurotransmitters and GPCRs, and PKA can also phosphorylate numerous distinct target proteins within the same cell. As a consequence, the cell can potentially respond to a rise in cAMP with a multitude of different, and sometimes opposing, effects[2]. Given the plethora of PKA targets within the same cell, the appropriate physiological response to a specific stimulus is achieved via compartmentalization of both GPCRs and the cAMP/PKA signal[3]. Compartmentalized signalling allows different GPCRs to generate unique spatially restricted cAMP pools[4] that in turn activate defined subsets of localized PKA[5]. Phosphodiesterases (PDEs), the enzymes that degrade cAMP, play a key role in the spatial regulation of cAMP propagation, and contribute to defining boundaries of individual cAMP pools[6,7]. Selective phosphorylation is often achieved through tethering of PKA adjacent to specific targets via A kinase anchoring proteins (AKAPs)[8].

Multiple drugs currently in use for the treatment of various conditions, including cardiac disease, target the cAMP/PKA pathway. Given the compartmentalization of cAMP signals, a detailed understanding of the actual organization, regulation and function of individual cAMP compartments may allow targeting individual (versus overall global cAMP in the bulk cytosol) cAMP pools, to yield greater therapeutic specificity[9]. However, the size and location of distinct cAMP domains, the amplitude and kinetics of the cAMP signal within each domain, as well as the specific functional role of individual domains remain largely unknown.

Although real-time imaging of [cAMP] using fluorescence resonance energy transfer (FRET)-based reporters has enhanced our understanding of compartmentalized cAMP signalling[4,10–12], major drawbacks have been the limited resolution of existing reporters and the difficulty in directly comparing cAMP signals detected at different intracellular sites[13]. We use here a combination of computational techniques to develop a novel cAMP FRET-based sensor that can be targeted to different macromolecular complexes, with equal cAMP sensitivities, to afford direct comparison of cAMP signals at multiple subcellular sites. Using this new tool we investigate local cAMP responses to catecholamine in cardiac myocytes. By targeting this novel sensor to key protein complexes that regulate excitation–contraction coupling (ECC) we find that physiologically relevant cAMP signals operate within the nanometer range. The cAMP signals generated upon β-AR activation show remarkable and unsuspected local heterogeneity, dictated by the activity of PDEs. Such local coordination of the cAMP signal maximizes β-AR-induced increases in contractility, but also mediates opposing effects on the myofilament response to $Ca^{2+}$. Our novel approach provides unprecedented accuracy and fidelity in the detection of local cAMP signalling with direct applicability to other cellular systems. We also offer original insight into the regulation of fundamental mechanisms of cardiac contractility with profound implications for the pathogenesis and treatment of heart disease.

## Results

**Limitations of targeted Epac1-camps reporters.** Our strategy to study local cAMP signalling with high resolution was to target multiple FRET reporters to distinct subcellular sites. We initially fused the widely used cAMP FRET sensor Epac1-camps[14] to protein components of various multiprotein complexes. This sensor features the FRET pair Yellow Fluorescent Protein- Cyan Fluorescent Protein (YFP-CFP) at the amino- and carboxyl-termini of the cyclic nucleotide binding domain (CNBD) of the protein Epac1 (Fig. 1a). When expressed in CHO cells and challenged with a saturating stimulus (FRSK + IBMX) that should produce similar maximal responses at all sites, the targeted sensors exhibited variable maximal FRET changes. In the extreme case, no detectable FRET change was seen with PDE4A1-Epac1-camps, as previously reported[15] (Fig. 1b). Thus, these sensors are impractical for comparing cAMP signals at different targeted cellular sites.

**Design of a 'universal' FRET tag for cAMP.** To overcome the limitations of the Epac1-camps chimeras, we sought to design a FRET sensor based on a novel topology that could potentially serve as a 'universal' FRET moiety to tag proteins of interest without affecting FRET sensor properties. We investigated moving YFP from the amino terminus of the sensor to minimize potential interference from the targeting domain (TD) target interaction with the FRET module. As the cAMP-sensing moiety we chose the second CNBD of the regulatory subunit type IIβ of PKA (PKA-RIIβ) (Supplementary Fig. 1a). Sequence conservation analysis (Supplementary Fig. 1b) led us to predict that loop 4–5 within this CNBD may be a suitable point to insert the YFP with minimal risk of perturbing the folding of either the CNBD or the YFP (Fig. 1c). We named this novel sensor CUTie (cAMP Universal Tag for imaging experiments) to emphasize its versatility in fusion proteins. Coarse-grained molecular dynamics (MD) simulations of CUTie using the SIRAH force field[16] predicted that in the presence of cAMP the distance between CFP and YFP is 6.4 nm, resulting in an average FRET change of 19% upon ligand binding (Supplementary Fig. 1c and Supplementary Note 1).

**Generation and characterization of CUTie.** The construct for expression of CUTie (Fig. 1c) was assembled and the reporter expressed in CHO cells. Figure 1d shows a representative time course of CUTie FRET change upon application of a saturating stimulus. As predicted from MD simulations, and unlike Epac1-camps, where binding of cAMP results in decreased FRET[14], cAMP binding to CUTie increases FRET (Fig. 1d, inset). The sensitivity range and $EC_{50}$ for cAMP (Fig. 1e) were determined by measuring FRET changes in CHO cells expressing CUTie and in which intracellular [cAMP] was equilibrated with known [cAMP] in patch pipettes (after membrane rupture). Fusion of CUTie to a number of targeting proteins (Fig. 1f) did not significantly alter the kinetics or the maximal amplitude of the FRET change (Fig. 1g), including fusion to PDE4A1, which had abolished FRET when fused to Epac1-camps (Fig. 1b).

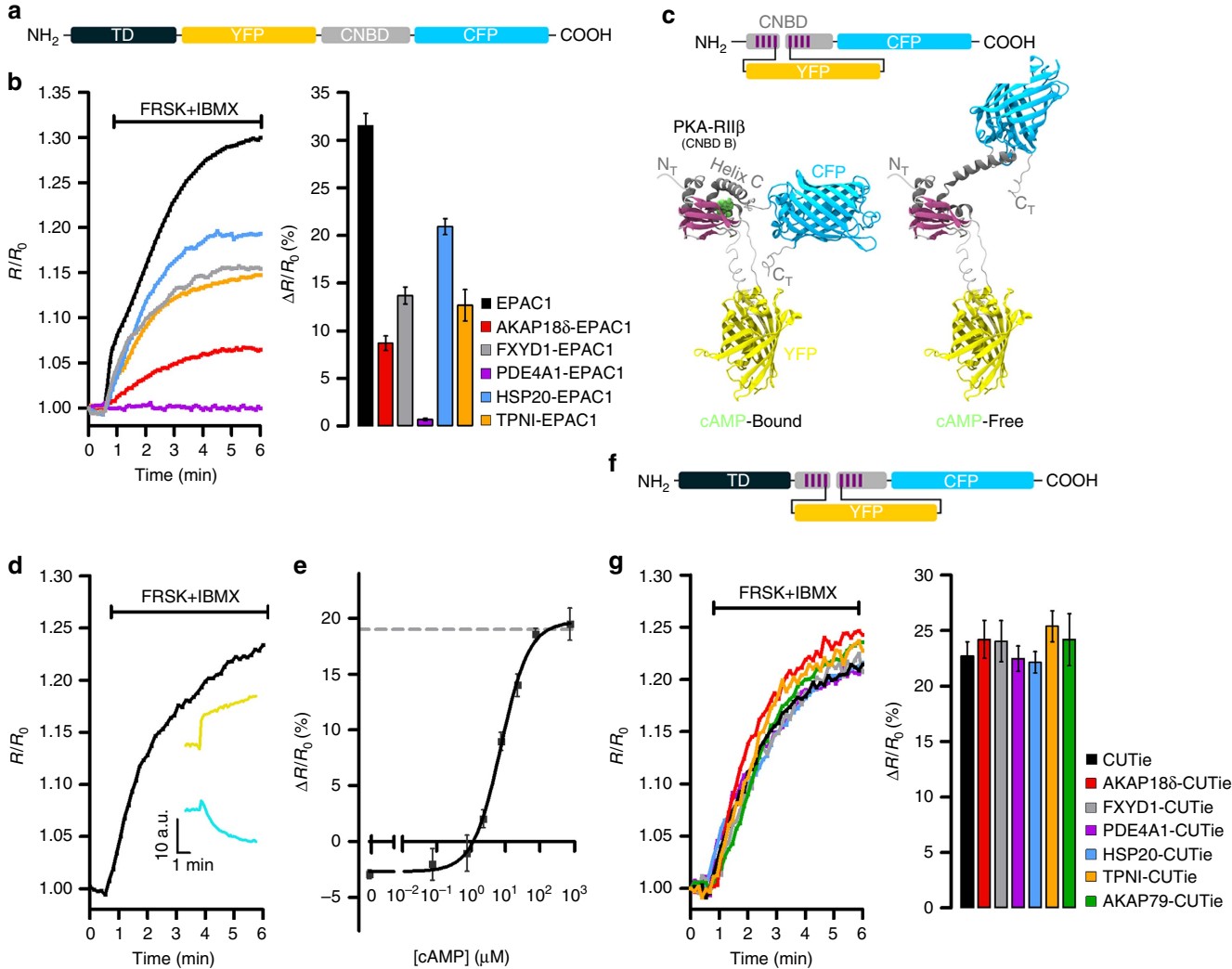

**Figure 1 | Generation of a universal FRET-tag for cAMP detection at specific macromolecular complexes.** (**a**) Schematic representation of the targeted Epac1-camps chimeras. CNBD, cyclic nucleotide binding domain; TD, targeting domain. (**b**) Representative kinetics of cAMP change (left panel) and summary of the experiments performed (right panel) in CHO cells expressing untargeted Epac1-camps (EPAC1) or its targeted versions upon application of the adenylyl cyclase activator forskolin (FRSK, 25 μM) and the phosphodiesterase inhibitor 3-isobutyl-1-methylxanthine (IBMX, 100 μM), a treatment that generates an intracellular amount of cAMP that saturates Epac1-cAMPs (ref. 27). AKAP18δ, A-kinase anchoring protein 18δ (ref. 17); FXYD1, phospholemman; HSP20, heat shock protein 20; TPNI, troponin I; PDE4A1, phosphodiesterase 4A1. Bars show FRET change at saturation. $N \geq 5$ from at least five independent experiments, all samples are significantly different from each other by one-way ANOVA and Bonferroni *post hoc* test, $P \leq 0.05$, except FXYD1-EPAC1 versus TPNI-EPAC1, which is not significant. (**c**) Top: schematic representation of CUTie. Bottom: ribbon representation of the predicted molecular structures of CUTie in its cAMP-bound or cAMP-free forms. cAMP is shown in green. (**d**) Representative kinetics of FRET change and corresponding CFP and YFP emission intensity curves (inset) recorded in a CHO cell expressing CUTie and treated with a saturating stimulus. (**e**) Concentration–response calibration curve generated using CHO cells expressing CUTie and microinfusion of known concentrations of cAMP via a patch pipette. $EC_{50} = 7.4$ μM, sensitivity range between 0.5 μM and 50 μM. Hill coefficient is 1.07. Broken line indicates maximal FRET change as predicted by MD simulations. (**f**) Schematic representation of CUTie chimeras. For each concentration point $N \geq 5$ from at least five independent experiments. (**g**) Representative kinetics of FRET change (left panel) and mean maximal FRET change (right panel) recorded at saturation. $N \geq 10$ from three independent experiments. One-way ANOVA analysis with Bonferroni's *post hoc* correction shows no significant difference between all samples. AKAP79, A-kinase anchoring protein 79. All values are means ± s.e.m.

**Expression of targeted CUTie in cardiac myocytes.** When expressed in neonatal rat ventricular myocytes (NRVM) the targeted CUTie reporters showed the expected specific localization (Supplementary Fig. 2a). For further studies here we focused on three key multiprotein complexes involved in ECC: the AKAP18δ/SERCA/PLB complex localized at the SR, regulating $Ca^{2+}$ reuptake[17]; the myofilament-localized troponin complex (TPNI/TPNT/TPNC) that regulates myofilaments $Ca^{2+}$ sensitivity[18]; and AKAP79/β-AR/adenylyl cyclase/LTCC complex at the plasmalemma[19] that regulates cAMP synthesis[20]

and LTCC $Ca^{2+}$ influx[21]. Figure 2a shows the predicted localization of CUTie fused to AKAP18δ, TPNI and AKAP79. When expressed in adult rat ventricular myocytes (ARVM), correct localization of AKAP79-CUTie, AKAP18δ-CUTie and TPNI-CUTie was confirmed (Fig. 2b). Co-immunoprecipitation of GFP and western blotting analysis confirmed that the targeted sensors are part of the expected macromolecular complex in ARVM (Fig. 2c) as well as in NRVM (Supplementary Fig. 2b). Expression of targeted CUTie does not affect basal or β-AR-stimulated myocyte responses (Fig. 2d and Supplementary

Fig. 2d,e). Critically, targeting of CUTie chimeras to myocyte subcellular compartments does not affect the kinetics or the maximal FRET change, both at saturating (Fig. 2e) and sub-saturating cAMP concentrations (Supplementary Fig 2c–e). The *in-cell* cAMP concentration–response curves for AKAP79-CUTie, AKAP18δ-CUTie and TPNI-CUTie are superimposable (Fig. 2f), confirming equivalent cAMP sensitivity and dynamic range. For AKAP79-CUTie EC$_{50}$ is 7.17 µM and Hill coefficient is 1.23; for AKAP18δ-CUTie EC$_{50}$ is 7.14 µM and Hill coefficient is 1.04; for TPNI-CUTie EC$_{50}$ is 7.24 µM and Hill coefficient is 1.15. The

sensors can therefore be used for direct comparison of cAMP changes at the sites where they localize.

**Differential local [cAMP] during β-AR activation.** We next tested cAMP responses to β-AR stimulation by isoproterenol (ISO) in ARVM expressing AKAP79-CUTie, AKAP18δ-CUTie or TPNI-CUTie. Figure 3a–c shows that ISO (5 nM) induced significantly smaller and more delayed cAMP response at TPNI than at AKAP18δ or AKAP79. Such heterogeneity in the cAMP

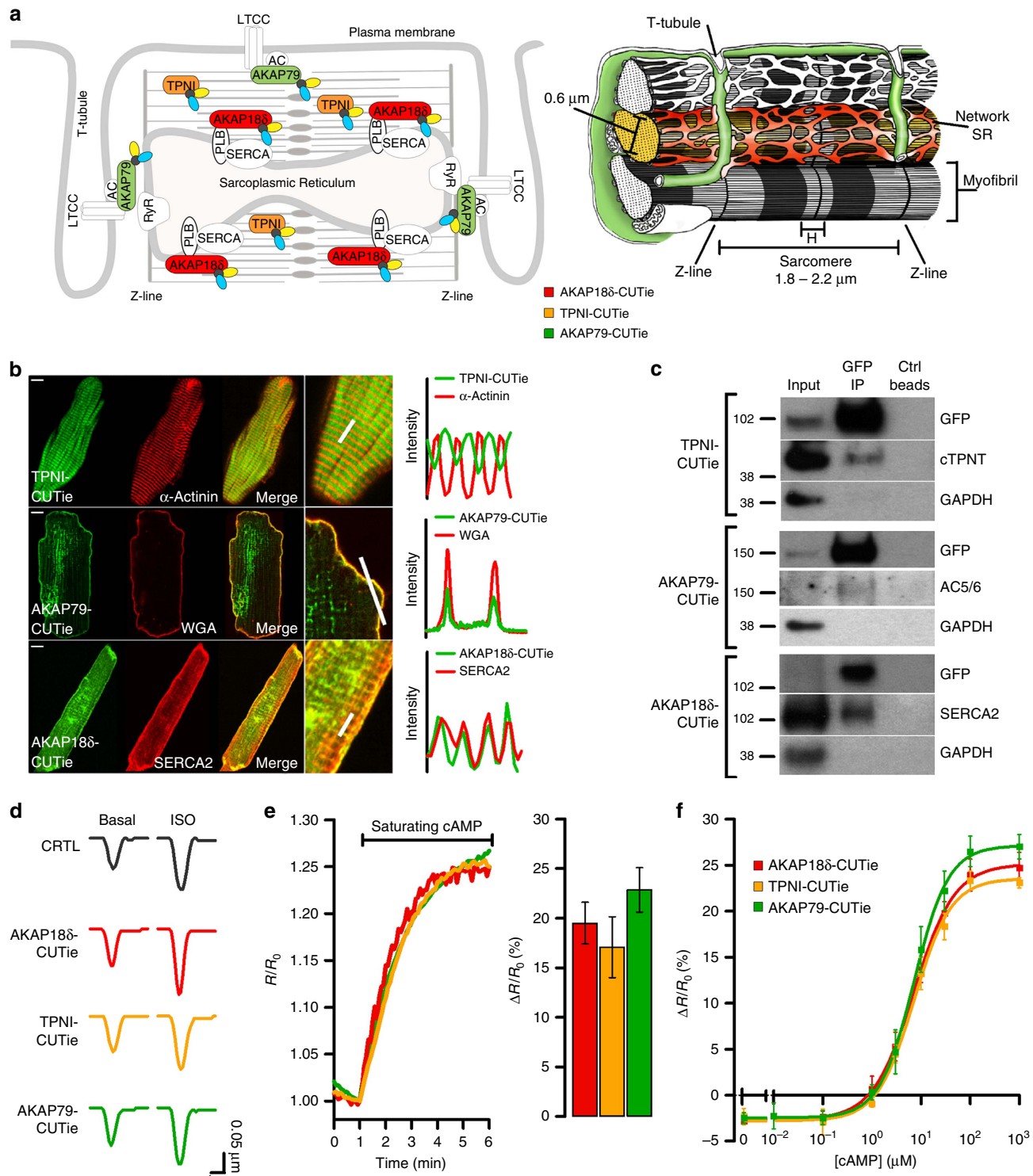

response was detected over a range of ISO concentrations, including at saturating concentrations (Supplementary Fig. 3). Similar results were obtained in NRVM (Supplementary Fig. 4a–c). The time course of ISO-dependent phosphorylation of TPNI and PLB reflects the different kinetics of the cAMP response (Fig. 3d). Notably, cAMP responses were synchronous and of similar amplitude at the three sites when ISO was applied in the presence of the PDE inhibitor IBMX (Fig. 3e–g and Supplementary Fig. 4d–f) or the cells were treated with IBMX alone (Supplementary Fig. 4d–f). Western blot analysis confirmed a faster and increased phosphorylation of TPNI (and comparable to PLB) upon IBMX treatment (Fig. 3h). Thus PDE activity is most strongly limiting [cAMP] at TPNI versus SR or plasmalemmal sites.

**Maximal stimulated inotropy requires cAMP nanodomains.** PKA-dependent phosphorylation of LTCC and PLB causes larger amplitudes of $Ca^{2+}$ transient and contraction, whereas PKA-mediated TPNI phosphorylation reduces myofilament $Ca^{2+}$ sensitivity and limits contraction amplitude. Both PLB and TPNI phosphorylation contribute to speed up relaxation during β-AR stimulation[1]. We hypothesized that differential regulation of cAMP at these sites may be required to optimally coordinate PKA-dependent phosphorylation to achieve maximal enhancement of contraction and relaxation. To test this hypothesis, we measured fractional shortening of ARVM in response to cAMP generated by ISO (compartmentalized signalling) or IBMX (homogeneous signalling) and measured simultaneously in individual ARVM the bulk cytosolic cAMP signal (assessed using the cytosolic, untargeted FRET reporter EPAC-S[H187] (ref. 22)). We found that application of 5 nM ISO results in significantly larger increase in fractional shortening as compared to 100 μM IBMX (Supplementary Fig. 5a–e). We then used a concentration of ISO (0.3 nM) that elicits a global cAMP response, as detected by EPAC-S[H187], similar to that resulting from inhibition of PDEs with 100 μM IBMX (Fig. 4a, top panel inset). However, unlike 100 μM IBMX, which generates a clearly detectable and equal FRET change at the three sites (Supplementary Fig. 6b), 0.3 nM ISO generates a response that is below the detection limit of the targeted sensors (Supplementary Fig. 6a). When we measured simultaneously in individual ARVM the bulk cytosolic cAMP signal and sarcomere shortening on application of 0.3 nM ISO (Fig. 4a) or 100 μM IBMX (Fig. 4b), we found that 0.3 nM ISO increased contractility significantly more than 100 μM IBMX (Fig. 4a–d and

Supplementary Fig. 6c). Notably, the blunted inotropic effect of IBMX is not due to a weaker $Ca^{2+}$ response, because IBMX did not limit the rise in $Ca^{2+}$ transient amplitude versus ISO (Fig. 4e,f and Supplementary Fig. 6d). Thus, with ISO we observed a greater contractile benefit for the same $Ca^{2+}$ enhancement. That is consistent with lower local [cAMP] at TPNI, weaker TPNI phosphorylation and higher myofilament $Ca^{2+}$ sensitivity with ISO.

**Mathematical modelling of local cAMP signalling.** To gain additional mechanistic insight regarding differential myofilament phosphorylation effects, we used a detailed cardiac myocyte model of $Ca^{2+}$, ionic currents and contraction, which includes cAMP-PKA-dependent signalling at nine known PKA targets (including three myofilament targets, TPNI, MyBPC and titin). This was obtained by incorporating our recent contractile model[23] into our mouse ventricular myocyte model[24], to better mimic experimental $Ca^{2+}$, contraction and PKA targets in rats (Supplementary Fig. 7a and Supplementary Note 2). In the baseline model, cAMP rises uniformly at all PKA targets upon β-AR stimulation. This corresponds to the experimental case of PDE inhibition with IBMX, whereby cAMP rises similarly at the plasmalemma, SR and TPNI. Figure 5a, left shows that our model agrees qualitatively with experimental data, with $Ca^{2+}$ transient and contraction amplitudes rising with IBMX (versus control) by 114% and 106%, respectively. To mimic the weak TPNI-CUTie changes with ISO versus IBMX, we decreased the cAMP rise at the myofilaments by 50% (TPNI, Titin and MyBPC). However, this limited the increases in $Ca^{2+}$ transient and especially in contraction, compared to uniform [cAMP] (to 88% and 72%, respectively), contrary to experiments in Fig. 4 where ISO increased contraction by more than $Ca^{2+}$ transients.

The model allows us to test the effect of attenuating cAMP effects at any individual PKA target[23], so we attenuated all three myofilament targets individually and collectively (Supplementary Fig. 7b). The only condition that showed a greater increase in shortening versus $Ca^{2+}$ transients was when only the TPNI effect was reduced (myofilament $Ca^{2+}$ desensitization), but the MyBPC and titin effects responded fully (bar graphs in Fig. 5a and Supplementary Fig. 7b). Indeed, in that case we found good agreement between simulation and experiments, whereby ISO versus IBMX increased $Ca^{2+}$ transients less than contraction (88 versus 113%, respectively; Fig. 5a, right). Thus, even within the myofilaments there may be differential regulation of PKA-dependent phosphorylation at TPNI, MyBPC and titin.

**Figure 2 | Targeting of CUTie to cardiac myocyte subcellular sites.** (**a**) Schematic representation of a portion of a cardiac myocyte in 2-D (left) and 3-D (right). In the panel on the right three myofibrils are shown, for two of which the sarcoplasmic reticulum (SR) surrounding individual myofibrils is depicted. The plasmalemma and its invaginations in T-tubules, which run along the Z-lines delimiting individual sarcomeres, are also shown. Colour shaded areas indicate the expected localization of targeted reporters. For clarity, only representative areas have been shaded. AKAP79-CUTie (green) is targeted to the surface of the plasmalemma facing the intracellular space; AKAP18δ (red) is targeted to the network SR surrounding individual myofibrils where SERCA also localize; TPNI-CUTie (yellow) localizes to the myofilaments constituting individual myofibrils, with exclusion of the Z-lines and the H zone. α-actinin localizes to the Z-line. Dimensions are from (ref. 54), 3-D schematic representation modified from (ref. 55). (**b**) Confocal images of ARVM expressing TPNI-CUTie, AKAP79-CUTie or AKAP18δ-CUTie showing the localization of the sensor relative to the localization of α-actinin, wheat germ agglutinine (WGA) and SERCA2, respectively. Scale bar: 10 μm. A merge of two images and a magnified detail for each cell are also shown. On the right, the fluorescence intensity profiles for the indicated targeted sensors (green) and the reference signal (red) are shown. Intensities were calculated along the line indicated in the zoomed-in images. (**c**) Western blotting analysis of proteins in complex with targeted CUTie. ARVM were infected with adenovirus carrying TPNI-CUTie, AKAP18δ-CUTie or AKAP79-CUTie. Targeted CUTie chimeras and their interacting proteins were pulled down using GFP beads. Membranes were probed with the antibodies indicated on the right of each panel. (**d**) Representative kinetics of sarcomere shortening recorded in ARVM expressing the targeted CUTie sensors in basal conditions (left) and on application of ISO 1 nM (right). Representative of at least four biological replicates. (**e**) Representative kinetics of cAMP change recorded in NRVM microinfused with 1 mM cAMP and 100 μM IBMX via a glass pipette (left panel) and mean maximal FRET change ± s.e.m. at saturation for all measurements performed (right panel). $N \geq 10$ from three biological replicates. No significant difference by one-way ANOVA and Bonferroni's *post hoc* test. (**f**) Concentration–response calibration curves generated by microinfusion of known concentrations of cAMP in CHO cells stably expressing TPNI-CUTie, AKAP18δ-CUTie or AKAP79-CUTie. For each concentration point $N \geq 5$.

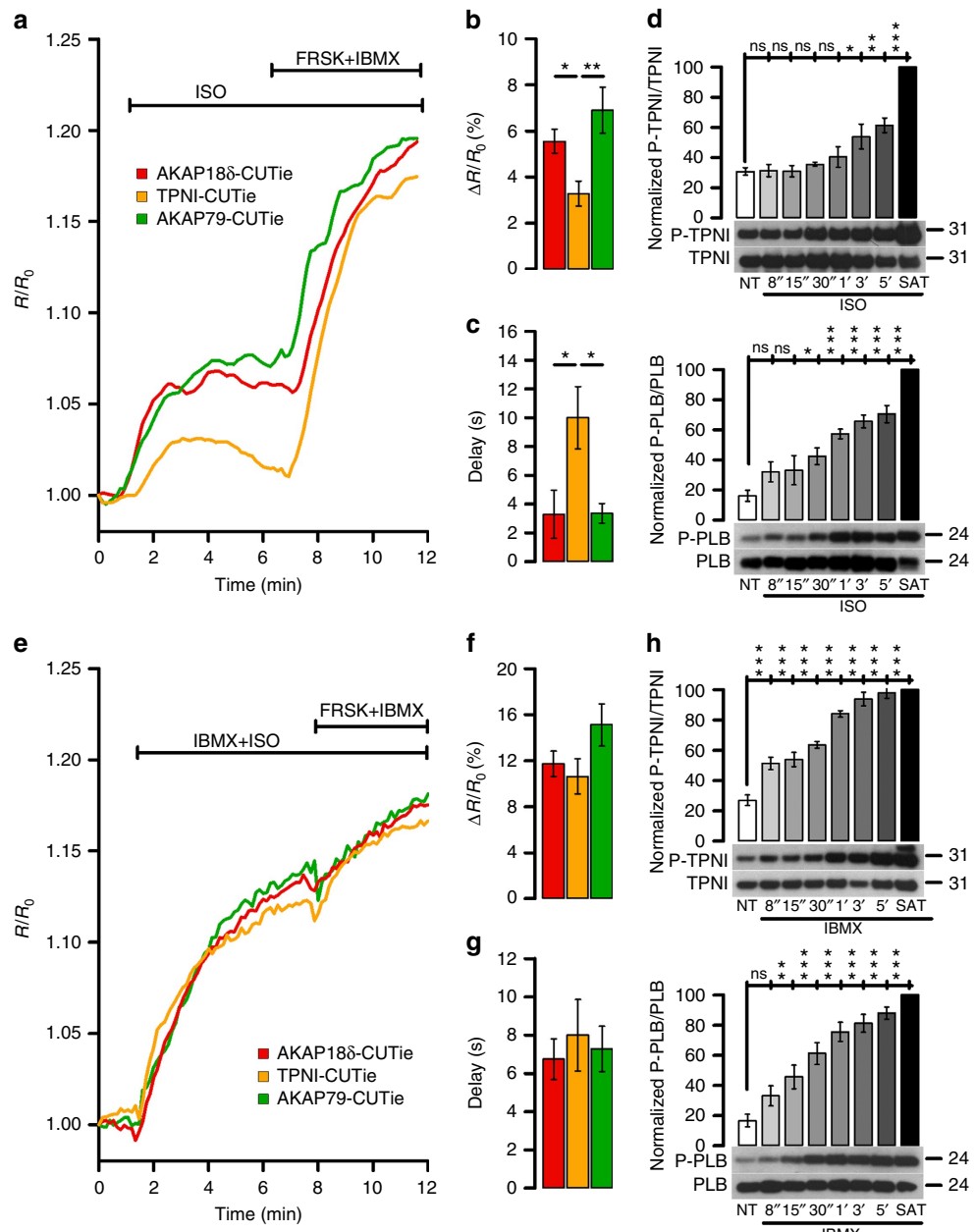

**Figure 3 | Differential regulation of the cAMP signal at different β-adrenergic targets.** (**a**) Representative kinetics and summary (**b**) of FRET change recorded in ARVM individually expressing AKAP18δ-CUTie, TPNI-CUTie or AKAP79-CUTie in response to bath application of 5 nM ISO. One-way ANOVA with Bonferroni's post-hoc correction. (**c**) Delay from application of 5 nM ISO to first time point at which a FRET change was detected for all experiments performed as in **a**. One-way ANOVA with Bonferroni's *post hoc* correction. (**d**) Time course of TPNI (top) and PLB (bottom) phosphorylation on application of 1 nM ISO to NRVMs. The level of phosphorylation on application of a saturating stimulus (SAT: 25 μM FRSK + 100 μM IBMX) for 10 min is also shown. NT, no treatment. $N = 5$ biological replicates. One-way ANOVA with Dunnett's *post hoc* correction (all columns versus NT column). (**e**) Representative kinetics and summary (**f**) of FRET change recorded in ARVM individually expressing AKAP18δ-CUTie, TPNI-CUTie or AKAP79-CUTie in response to bath application of 1 nM ISO in combination with 100 μM IBMX. One-way ANOVA with Bonferroni's *post hoc* correction. (**g**) Delay from application of 1 nM ISO + 100 μM IBMX to first time point at which a FRET change was detected for all experiments performed as in **e**. (**h**) Time course of TPNI (top) and PLB (bottom) phosphorylation on application of 100 μM IBMX to NRVM. $N = 5$ biological replicates. One-way ANOVA with Dunnett's *post hoc* correction (all columns versus NT column). Bars indicate means ± s.e.m. For (**b,c,f,g**) $N \geq 7$ from at least $N = 5$ biological replicates. *$P \leq 0.05$, **$P \leq 0.01$, ***$P \leq 0.001$.

To test this prediction experimentally, we measured differences in phosphorylation of TPNI, PLB and MyBPC with ISO versus IBMX (Fig. 5b–d). We also used PKA inhibition (H89, 30 μM) and saturating activation to estimate minimal and maximal phosphorylation levels. Addition of 0.3 nM ISO robustly increased phosphorylation of both MyBPC and PLB, but not

TPNI (Fig. 5b–d). In contrast, raising [cAMP] globally with 100 μM IBMX resulted in strong phosphorylation for all three targets. Together with the data at 5 nM ISO (Fig. 3a,b,e,f), these results further confirm that ISO stimulation generates compartmentalized cAMP signals and differential PKA-mediated phosphorylation, with a blunted cAMP/PKA signal at TPNI compared

to the other sites. The data strongly support the conclusion that a smaller compartmentalized cAMP signal (as generated by 0.3 nM ISO) is more effective in enhancing inotropy than a significantly larger, homogeneous increase in cAMP (as generated by 100 μM

IBMX). The data also confirm the model predictions that limiting TPNI phosphorylation allows greater increase in contraction for a smaller increase in $Ca^{2+}$ transients. The more gradual recruitment of TPNI phosphorylation at higher levels of β-AR activation

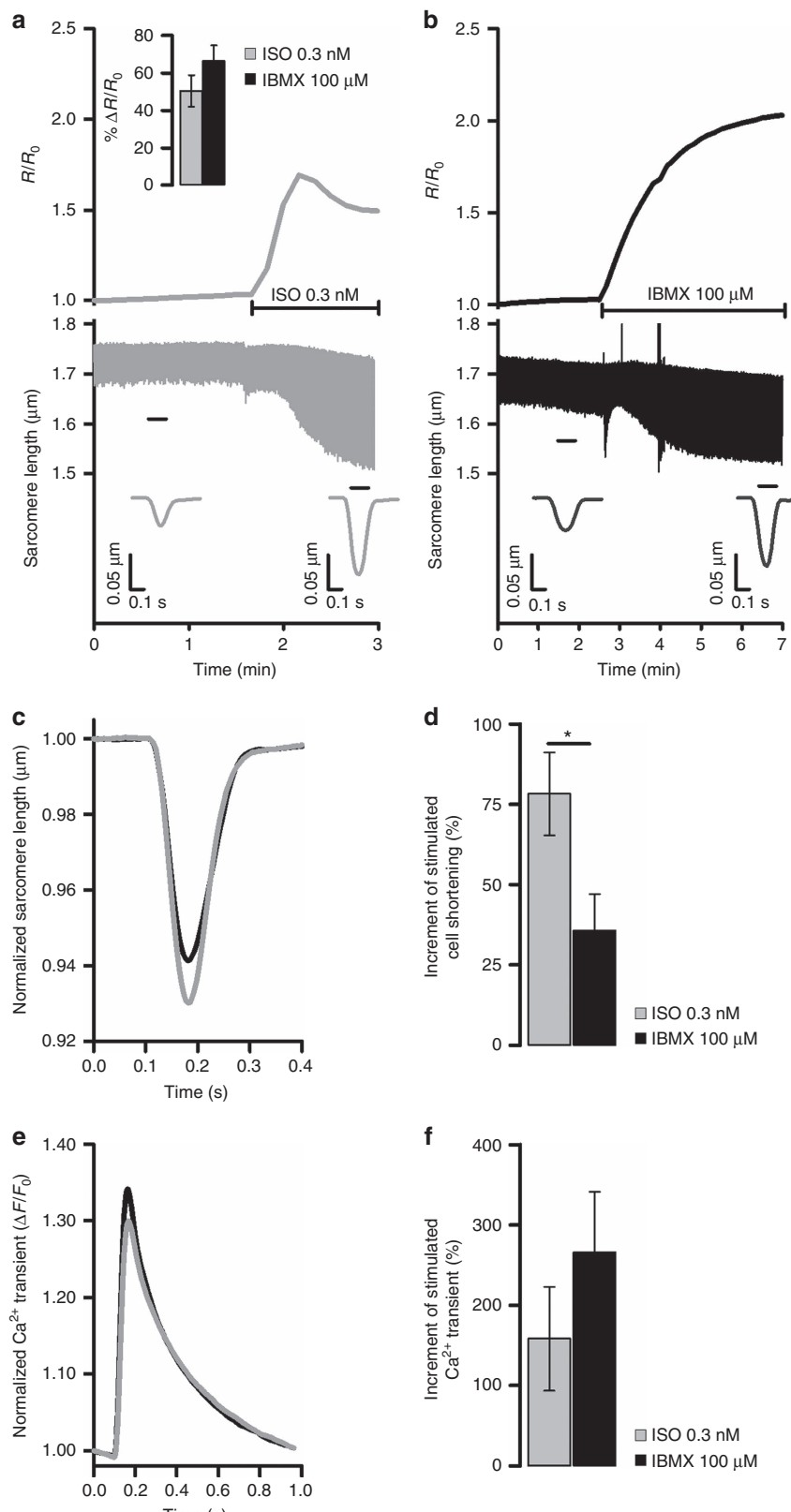

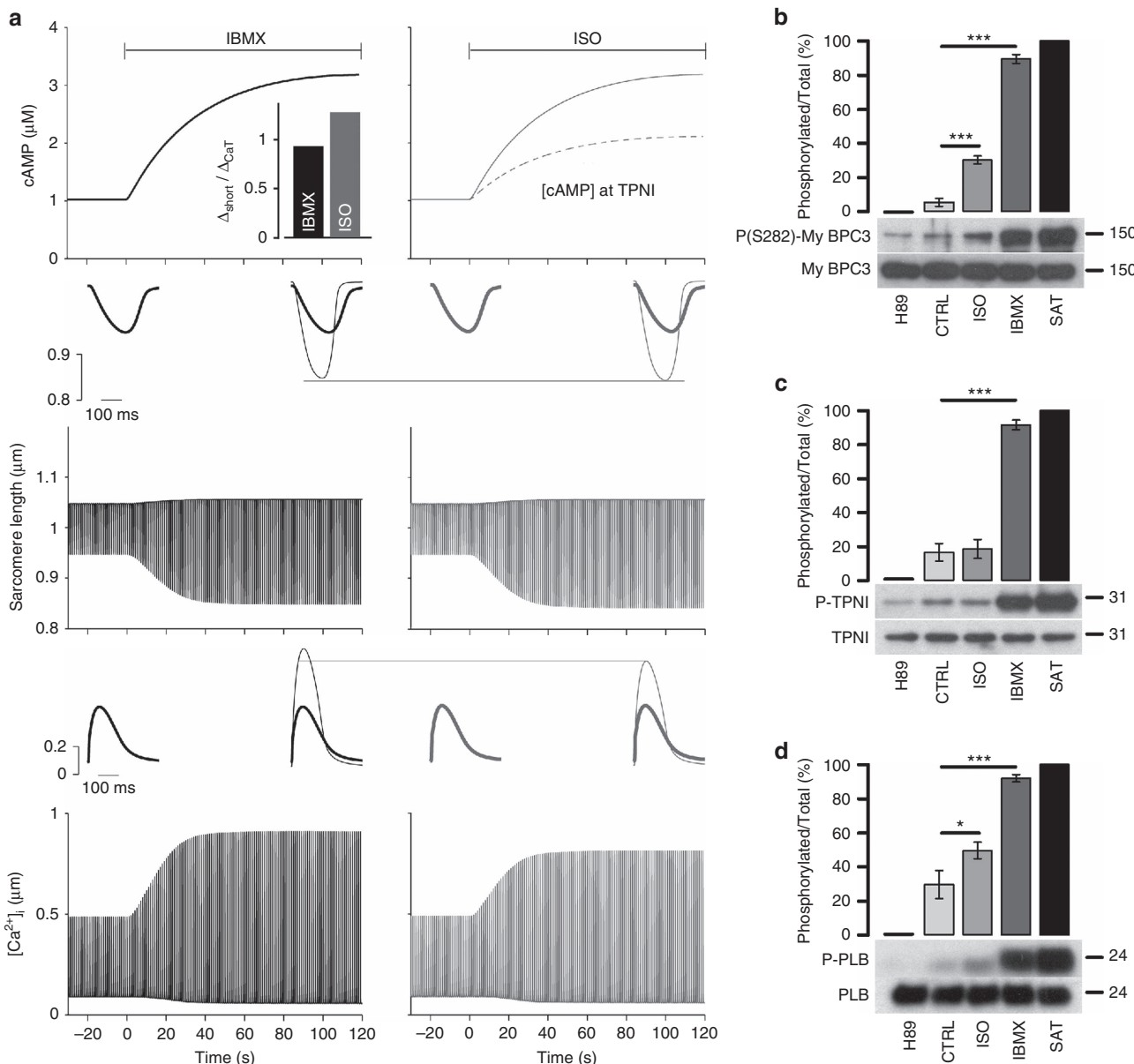

**Figure 5 | Mathematical model of compartmentalized cAMP signalling.** (**a**) Simulated cAMP rise (top), myocyte shortening (middle) and $Ca^{2+}$ transient (bottom) upon administration of IBMX or ISO at time = 0 s. Inset at the left shows baseline condition, and inset at right shows overlapping baseline condition (grey) and steady-state response to cAMP increase. Bar graph in **a** shows the ratio of the increase in maximal shortening to increase in $Ca^{2+}$ transient amplitude induced by IBMX and ISO. (**b**) Western blotting analysis of cell lysates obtained from ARVM treated with H89 (30 µM), vehicle (CTRL), ISO (0.3 nM), IBMX (100 µM) or 25 µM FRSK + 100 µM IBMX (SAT) for 10 min. Membranes were probed for total and phosphorylated MyBPC (**b**), TPNI (**c**) and PLB (**d**). Graphs show densitometric analysis and present the ratio value of phosphorylated to total protein expressed as percentage after normalization to H89 treatment (taken as zero) and to maximal phosphorylation at saturation (taken as 100%). Values are means ± s.e.m. *$P \leq 0.05$, ***$P \leq 0.001$. $N \geq 8$ independent experiments from at least five biological replicates. One-way ANOVA with Dunnett's *post hoc* correction.

**Figure 4 | Differential local regulation of cAMP signals is necessary for maximal stimulated inotropy.** (**a**) Representative time course of global cytosolic cAMP change (top) and sarcomere shortening (bottom) recorded simultaneously in the same ARVM expressing the cytosolic FRET reporter EPAC-S[H187] on application of 0.3 nM ISO or (**b**) 100 µM IBMX. Inset at the top of **a** shows mean FRET change measured in ARVM expressing the cytosolic FRET reporter EPAC-S[H187] on application of 0.3 nM ISO or 100 µM IBMX. Bars are means ± s.e.m., no significant difference by unpaired *t*-test. In **a,b** cells were paced at 1 Hz. Inserts at the bottom of **a,b** indicate sarcomere shortening kinetics averaged over the time interval indicated by the black bar. (**c**) Normalized mean sarcomere shortening kinetics measured at steady state after the application of ISO (0.3 nM) or IBMX (100 µM), as indicated. (**d**) Effect of 0.3 nM ISO or 100 µM IBMX on sarcomere shortening measured in all experiments as shown in **a,b**. Shortening is expressed as percent increment over control (before the stimulus) calculated as ($\Delta$ shortening/shortening$_{control}$) × 100, where $\Delta$ shortening = (shortening$_{stimulated}$ − shortening$_{control}$). Unpaired *t*-test. (**e**) Averaged normalized $Ca^{2+}$ transient recorded on application of 0.3 nM ISO or 100 µM IBMX. $N \geq 6$. (**f**) Effect of 0.3 nM ISO or 100 µM IBMX on the amplitude of the $Ca^{2+}$ transient expressed as percent increase over control (before the stimulus). Unpaired *t*-test shows no significant difference. Bars are means ± s.e.m. *$P \leq 0.05$. For all experiments $N \geq 6$ from at least three biological replicates.

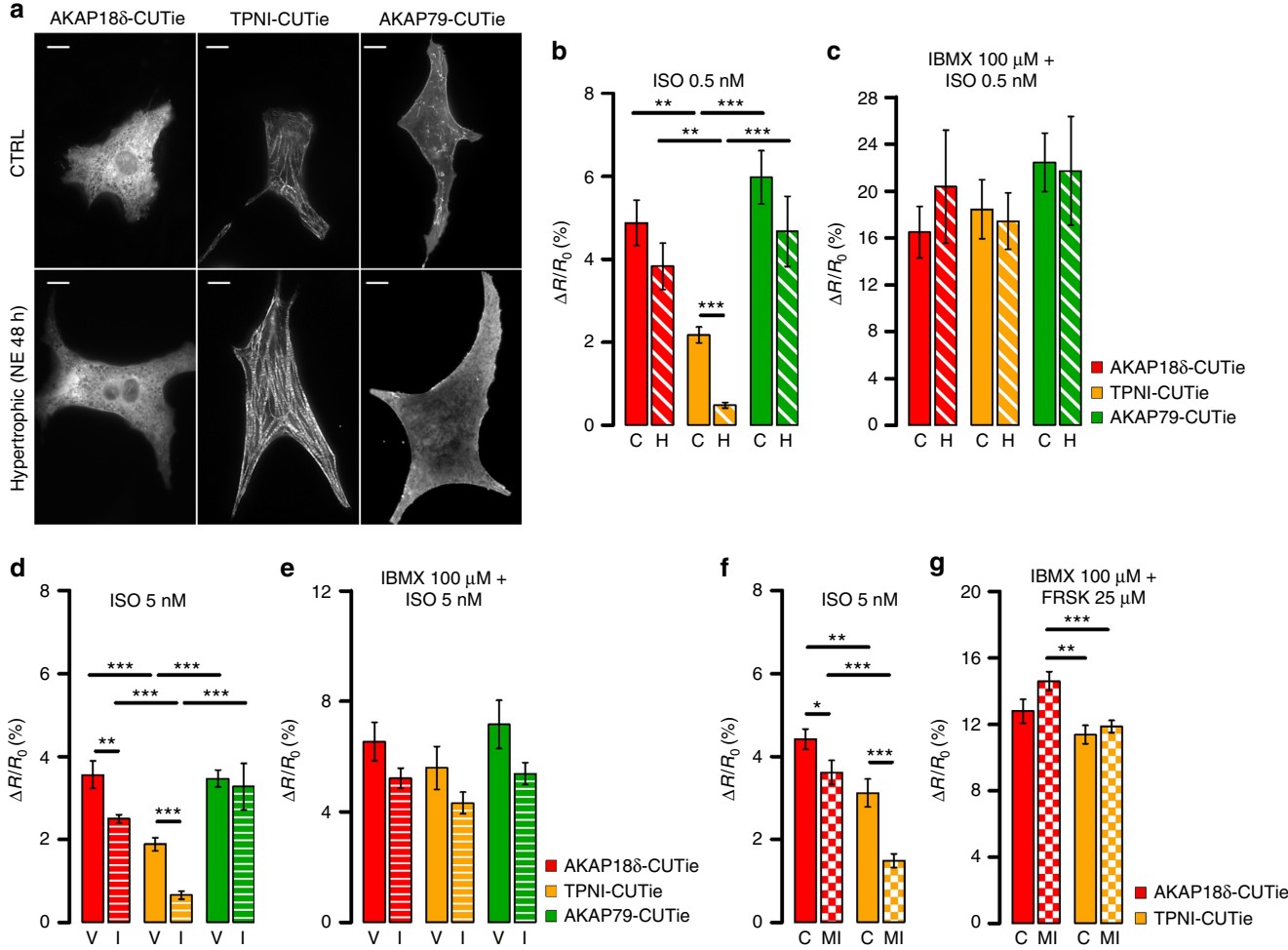

**Figure 6 | Compartmentalized cAMP signalling in hypertrophic cells.** (**a**) Localization of the targeted CUTie reporters in control (top) and hypertrophic (bottom) NRVM. Scale bar: 10 μm. Average cell size for control myocytes was $351.8 \pm 7.3\,\mu m^2$ and for hypertrophic myocytes $674.9 \pm 14.8\,\mu m^2$ ($N \geq 216$ cells from five biological replicates, $P \leq 0.001$). (**b**) Mean FRET change measured with targeted CUTie reporters in control (C) and hypertrophic (H) NRVM on application of 0.5 nM ISO and (**c**) 0.5 nM ISO in the presence of 100 μM IBMX. $N \geq 6$ from five biological replicates. (**d**) Mean FRET change measured with targeted CUTie reporters in isolated ARVM from minipump vehicle- (V) and ISO-infused (I) rats, on application of 5 nM ISO and (**e**) 5 nM ISO in the presence of 100 μM IBMX. $N \geq 8$ from at least six biological replicates. Bars indicate means ± s.e.m. Unpaired $t$-test. (**f**) Mean FRET change measured with targeted CUTie reporters in isolated ARVM from age-matched control (C) and rats subjected to myocardial infarction (MI) at 16 weeks after coronary artery ligation, on application of 5 nM ISO and (**e**) saturating stimulus. $N \geq 16$ from at least seven biological replicates. In all graphs bars indicate means ± s.e.m. Unpaired $t$-test applied for comparison between treatment groups, one-way ANOVA and Bonferroni's *post hoc* test for comparison among sensors. *$P \leq 0.05$, **$P \leq 0.01$, ***$P \leq 0.001$.

may aid relaxation and assure diastolic relaxation under high sympathetic tone, where higher heart rates limit diastolic time.

**TPNI is selectively vulnerable to reduced adrenergic input.** Given the reduced [cAMP] rise at TPNI versus PLB on ISO stimulation, we asked whether this subcellular compartment may be particularly vulnerable in conditions such as cardiac hypertrophy and heart failure (HF) where the cAMP response to catecholamine is blunted as a consequence of adrenergic desensitization[25]. To address this question, we expressed the targeted CUTie sensors in NRVM treated for 48 h with 10 μM norepinephrine, a well-established *in vitro* model of cardiac hypertrophy (Fig. 6a)[26]. As shown in Supplementary Fig. 8b the *in vitro* hypertrophied myocytes exhibit an attenuated global cAMP response to ISO. Detection of [cAMP] with the targeted reporters revealed that whereas the response to 0.5 nM ISO at TPNI is significant in control cells it is almost undetectable in

hypertrophic cells (Fig. 6b). Unexpectedly, we found that the cAMP signal at AKAP79 and AKAP18δ was not significantly different in control versus hypertrophic cells (Fig. 6b), despite global cAMP being significantly reduced (Supplementary Fig. 8b). Application of 0.5 nM ISO in the presence of 100 μM IBMX abolished any difference between subcellular compartments and between control and hypertrophic cells (Fig. 6c).

We next probed the local cAMP response in hypertrophic myocytes from rats subjected to minipump infusion of ISO for 7 days or failing myocytes from rats subjected to myocardial infarction (MI) upon coronary artery ligation (the characterization of these two *in vivo* models is summarized in Supplementary Table 1). As shown in Fig. 6d and f, respectively, both ISO infusion and MI result in a dramatic decrease in the cAMP response at TPNI compared to the other sites. Interestingly, in both *in vivo* models, the response at AKAP18δ was also significantly reduced compared to control cells, albeit to a lesser extent (Fig. 6d,f), suggesting that additional mechanisms altering

**Table 1 | cAMP concentration and PKA activity in different compartments of healthy and hypertrophic myocytes.**

| Stimuli | Control | | | | | | Hypertrophic | |
|---|---|---|---|---|---|---|---|---|
| | 1 nM ISO | | 0.5 nM ISO | | 100 μM IBMX | | 0.5 nM ISO | |
| | cAMP | PKA | cAMP | PKA | cAMP | PKA | cAMP | PKA |
| *Compartments* | | | | | | | | |
| AKAP18δ | 5.8 | 45 | 3 | 13 | 5.1 | 37 | 2.6 | 9 |
| TPNI | 3.7 | 21 | 2.1 | 5 | 5 | 36 | 1.4 | 0 |
| AKAP79 | 5.4 | 40 | 3.3 | 16 | 4 | 35 | 2.8 | 11 |
| Bulk cytosol | 4.1 | 26 | 2.6 | 9 | 3 | 14 | 1.6 | 2 |

Concentration of cAMP (μM) and increase in PKA activity over basal (expressed as percent of maximal activation) in untreated (control) and NE-treated (hypertrophic) NRVM at AKAP18δ, TPNI, AKAP79 and in the bulk cytosol. cAMP concentrations are calculated from FRET changes as measured with AKAP18δ-CUTie, TPNI-CUTie, AKAP79-CUTie or CUTie using the calibration curves shown in Figs 1e and 2f. FRET changes to calculate cAMP concentrations are from Supplementary Fig. 4b,e and Fig. 6b,c. For completeness, data for cAMP changes in the bulk cytosol in response to 1 nM ISO and 100 μM IBMX are also shown. Increase in PKA activity is calculated from cAMP concentrations derived as above using the PKA activity calibration curve shown in Supplementary Fig. 9. Basal PKA activity is set at 3% based on the *in-cell* PKA activity calibration curve.

the regulation of local cAMP signalling at the AKAP18δ/PLB/SERCA complex may come into play at later stages during the myocardial remodelling process.

Using the concentration–response curves for cytosolic (Fig. 1e) and targeted (Fig. 2f) CUTie in combination with an *in-cell* calibration curve for PKA activation (Supplementary Fig. 9) we can estimate the impact on PKA activity of the cAMP signal generated at the sites studied. As summarized in Table 1, the difference between the cAMP response and the resulting PKA activation at TPNI and at the other compartments is exacerbated by reduced adrenergic input. Importantly, while 0.5 nM ISO applied to healthy NRVM results in significant activation of PKA, it does not generate sufficient cAMP to activate PKA at TPNI in hypertrophic cells. The same stimulus, however, generates comparable activation of PKA at other sites both in healthy and hypertrophic cells.

## Discussion
We developed a new FRET-based sensor, CUTie, that affords unprecedented fidelity and accuracy of detection of compartmentalized cAMP by quantitatively reporting and allowing direct comparison of cAMP levels in the environment immediately surrounding macromolecular complexes. This sensor offers a general method for fine mapping of cAMP signals in any cellular system. Although CUTie EC$_{50}$ for cAMP is higher than for other available cAMP FRET reporters[27], its sensitivity range is within physiological intracellular cAMP concentrations and adequate to measure cAMP changes elicited by sub-nanomolar β-adrenoceptor agonist.

We investigate the cardiac response to catecholamines and focus on three complexes that regulate ECC: the AKAP18δ/SERCA/PLB complex at the SR, the AKAP79/β-AR/adenylyl cyclase/LTCC complex at the plasmalemma and the troponin complex at the myofilaments. We show that the cAMP signal generated in response to β-AR stimulation differs at these three sites in both amplitude and kinetics. Such heterogeneity does not depend on the distance from the site of cAMP synthesis at the plasmalemma, but relies on differential local cAMP degradation by PDEs, as equal responses at the three sites are detected on PDEs inhibition. Given the localization of the complexes involved and the geometry of ARVM (Fig. 2a), an upper bound for the dimension of the cAMP domains investigated here can be estimated at 300 nm (based on the diameter of myofibrillar bundles), at least one order of magnitude smaller than previously thought[4]. Despite the small size of local cAMP nanodomains, the plasmalemma (T-tubules) and free SR network, which are up to 1.1 μm apart, exhibit similar local cAMP responses at AKAP79 and AKAP18δ. This suggests similar local cAMP tuning in these regions (at least under normal conditions). In contrast, our results suggest that differences in local regulation of cAMP may occur within tens of nanometers within the myofilament structure, between TPNI and MyBPC sites that are in very close physical proximity.

We observe that different treatments can generate comparable increases in bulk cytosolic cAMP (Fig. 4a, top panel inset) while resulting in significantly different cAMP levels in the environment immediately surrounding individual macromolecular complexes (Fig. 3a and Supplementary Fig. 4a,b). In addition, we find that some specific sites (AKAP79 and AKAP18δ), but not others (TPNI), can exhibit robust increases in cAMP level even with reduced adrenergic input (Fig. 6 and Table 1). Taken together these findings indicate that global and local cAMP are differentially regulated, and support a model whereby cytosolic cAMP ('bulk cAMP') is less involved than are very local cAMP nano-domains ('physiologically relevant cAMP') in regulating function.

We show that maximal contractility is achieved when the cAMP increase at TPNI is delayed and attenuated compared to the cAMP response at AKAP79 and AKAP18δ. Both mathematical modelling and experimental evidence indicate that optimal inotropy is achieved if the attenuated PKA-mediated phosphorylation at the myofilaments selectively affects TPNI while the level of phosphorylation of MyBPC is maintained high. Interestingly, differential phosphorylation of TPNI and MyBPC upon catecholamine stimulation was previously reported[28,29] although the mechanism remained unclear. PKA activity reporters targeted to plasmalemma, PLB and troponin T also recently revealed domain differences in PKA activity, particularly at the myofilaments[30]. Our findings here are compatible with a model whereby TPNI and MyBPC are under the control of distinct cAMP/PKA nano-domains, although it is also possible that other kinases (for example, CaMKII, which also phosphorylates MyBPC at S282 (ref. 31)) or reduced local phosphatase activity may be responsible for the stronger phosphorylation of MyBPC. It should be noted that TPNI contributes significantly to lusitropy during isometric contractions[32] and therefore our measurements of unloaded myocyte shortening are likely to underestimate the *in vivo* effect of TPNI phosphorylation on relaxation kinetics.

PDE inhibitors are approved drugs for the treatment of HF, where they are intended to correct defective cAMP signalling[33]. Although administration of PDE inhibitors can promote short-term inotropic responses, long-term functional improvement is not sustained and their administration is associated with increased mortality[34]. This has limited their utility to treatment of HF that is not responding to alternative therapies. Our findings may explain the long-term failure of this therapeutic approach, as PDE

inhibition, by obliterating the difference between cAMP nano-domains and reducing efficiency of contraction, effectively curtails beneficial local cAMP controls to optimize contractility.

Here we demonstrate that the spatial heterogeneity of the cAMP signal is accentuated in pathological stress conditions (chronic β-adrenergic activation or post-MI), where acute β-adrenergic induced [cAMP] elevation (within 2–6 min) is nearly abolished at TPNI, but well-preserved at plasmalemma targets (AKAP79) and only slightly limited at the SR (AKAP18δ). It is well established that PKA-mediated phosphorylation of TPNI reduces myofilament $Ca^{2+}$ sensitivity[35] and data support a role for $Ca^{2+}$ sensitization in diastolic dysfunction[36], focal energy deficits[37] and arrhythmogenesis[38]. In addition, genetic models with unphosphorylatable TPNI show enhanced susceptibility to HF development under stress[39–41], implicating an increase in TPNI $Ca^{2+}$ sensitivity as a cause of contractile abnormality in the failing heart. Numerous studies have reported a significant decrease in TPNI phosphorylation in failing hearts versus control, both in animal models and in humans[42,43], a feature that has been shown to be associated with preserved PLB phosphorylation[44–46]. β-Adrenergic downregulation and/or upregulation of phosphatase activity[47] or delocalization and degradation of local PKA as a consequence of the remodelling process[44] have been suggested as possible mechanisms responsible for defective TPNI phosphorylation. Our study demonstrates an intrinsic propensity of the myocyte to $Ca^{2+}$ sensitization, as TPNI is selectively vulnerable to low phosphorylation as a consequence of attenuated local cAMP signalling. Thus, maximal contractility in response to stress is achieved at the cost of increased tendency for myofilament $Ca^{2+}$ sensitization and diastolic dysfunction. Our findings may thus explain the limited success of β-blocker therapy in HF with preserved ejection fraction[48] and suggest that therapeutic interventions aimed at normalizing $Ca^{2+}$ sensitivity may prove beneficial in the treatment of HF, particularly in the early stages of the pathological process. As local domains of cAMP are regulated by the activity of specific PDE isoforms[9], identification of the PDE isoform(s) involved in attenuation of cAMP signals at TPNI may provide a novel therapeutic target.

## Methods

**Generation of CUTie FRET sensor.** The details of the rational design of the CUTie sensor along with the computational methods applied to define the final protein sequence are provided in the Supplementary material section. A DNA fragment encompassing nucleotides 817–986 of the *Rattus norvegicus* protein kinase, cAMP-dependent, regulatory subunit type II beta (Prkar2β) (NM_001030020.1) (PDB ID: 1CX4) was PCR-amplified from cDNA and inserted between NheI and AgeI restriction sites upstream of Enhanced Yellow Fluorescent Protein (EYFP) (PDB ID: 1QY0), and a fragment encompassing nucleotides 987–1235 was PCR-amplified and inserted between EcoRI and SalI restriction sites downstream EYFP in the pEYFPc1 vector (Clontech Laboratories, CA, USA). Enhanced Cyan Fluorescent Protein (ECFP) (PDB ID: 1QY0) including the stop codon was fused downstream of nucleotide 1235 of the Prkar2β, between the SalI and BamHI restriction sites.

**Generation of targeted reporters.** Full length AKAPs 18, isoform δ (AKAP18δ) (NM_001001801), FXYD domain containing ion transport regulator 1 (FXYD1) (NM_001278718.1), heat shock protein, alpha-crystallin-related B6 (HSP20) (BC068046.1), TPNI type 3 (cardiac) (TPNI) (BC099631), cAMP-dependent protein kinase (AKAP79) (M90359.1) and cyclic AMP-specific phosphodiesterase (PDE4A1) (L27062.1) were PCR-amplified to incorporate NheII and HindIII restriction sites at their 5′ and 3′ ends, respectively. The resulting fragments were individually cloned, in frame, into the NheII site 5′ to the CUTie sensor, or into the HindIII site 5′ to the EPAC1-based FRET sensor[14]. Clones were screened for correct orientation of insert and selected clones were sequenced. The TPNI- and HSP20-targeted EPAC1 sensors were kindly provided by Prof G. Baillie, University of Glasgow, UK. Adenoviral vectors carrying the targeted reporters were generated by Vector BioLabs (Malvern, PA, USA).

**In-cell calibration by microinfusion.** A detailed protocol is described in ref. 49. In brief, mycoplasma free CHO (Hamster Chinese ovary) cells (from Sigma-Aldrich, UK) stably expressing the CUTie sensor were patch-clamped and monitored for FRET ratio change (see below) simultaneously. After establishing a tight seal between cellmembrane and patchpipette the whole-cell configuration was established. In this configuration, providing a direct access from the pipette to the cytoplasm, cAMP can either diffuse from the pipette into the cell or vice versa, depending on the cAMP concentration in the pipette solution. The FRET ratio changes elicited by a range of cAMP concentrations loaded into the patch pipette were recorded, analysed offline and computed to generate dose-response curves. All curves were corrected for quenching effects due to chloride in the pipette solution. Seal, whole-cell and access resistances were continuously monitored during the experiments. Seal resistances typically were in the range of several Gigaohm, whole-cell resistances were between 0.5 and 1 GΩ. Whole-cell access resistance was in the range of 80–100 nS, as computed online by WinWCP (Vers. 4.1.2, Strathclyde Electrophysiology Software, John Dempster, University of Strathclyde). Pipette solutions contained 20 mM NaCl, 140 mM KCl, 2 mM $MgCl_2$, 0.00404 mM $CaCl_2$, 0.1 mM BAPTA and 10 mM HEPES, supplemented with the required amount of cAMP-Na. For cardiomyocyte measurements KCl was replaced by 140 mM K-glutamate. All patch-pipette solutions were adjusted to match the intracellular pH. The extracellular solution contained 140 mM NaCl, 3 mM KCl, 3.2 mM $MgCl_2$, 2 mM $CaCl_2$, 15 mM glucose, 10 mM HEPES, and was adjusted with NaOH to pH 7.2. To prevent swelling or shrinkage artefacts, osmolality values of pipette and extracellular solutions were matched.

To determine the extent of PKA activation at different intracellular concentrations of cAMP a similar approach as described above was used, with a PKA-based FRET reporter[4] where the R subunit is tagged with CFP and the C subunit is tagged with YFP (see Supplementary Fig. 9a). On binding of cAMP to the R-CFP subunit the C-YFP subunit dissociates and FRET is abolished. It should be noted that this sensor displays identical sensitivity to cAMP-dependent activation as the native, untagged PKA[50] and can therefore be used to accurately measure PKA activation in the cell. The reporter was expressed in CHO cell and known concentrations of cAMP were delivered into the cells via a patch pipette while measuring FRET changes as described above.

**Isolation and culture of cardiomyocytes.** ARVM were isolated from 350 to 375 g male Sprague-Dawley rats as described[51]. Briefly, hearts were perfused on a Langendorff apparatus at 37 °C for 4–5 min through the coronary arteries with an oxygenated $Ca^{2+}$-free Tyrode Buffer solution (130 mM NaCl, 5 mM Hepes, 0.4 mM $NaH_2PO_4$, 5.6 mM KCl, 3.5 mM $MgCl_2$, 20 mM taurine, 10 mM glucose), and subsequently digested with 0.06 mg ml$^{-1}$ Liberase TH (Roche Diagnostics Limited, UK) in a 100 μM $Ca^{2+}$ containing Tyrode buffer solution for a further 15–20 min until the heart became flaccid. After perfusion, the left ventricle was removed, minced and resuspended in equal volume of 1% BSA $Ca^{2+}$-free Tyrode Buffer solution. Extracellular $Ca^{2+}$ was added incrementally to reach a final concentration of 1.0 mM. Cells were cultured in Minimum Essential Medium (MEM) (Sigma-Aldrich, UK), supplemented with 2.5% fetal bovine serum (FBS), 100 units per ml penicillin, 100 μg per ml streptomycin, 2 mM L-glutamine and 9 mM $NaHCO_3$ and plated on laminin (40 μg ml$^{-1}$) coated culture dishes. ARVM were left to adhere for 2 h in a 5% $CO_2$ atmosphere at 37 °C, before the medium was replaced with FBS-free MEM. Myocytes were infected for 3 h at multiplicity of infection of 10–100 with adenovirus encoding for FRET reporters. The medium was then replaced with fresh FBS-free MEM including 0.5 μM cytochalasin D (MP Biomedicals, UK). Infected cells were kept in culture for less than 36 h. Sprague-Dawley NRVM (2 days old) were cultured as described before[4]. For hypertrophy induction, cells were cultured in serum-free MEM for 24 h prior to addition of 10 μM norepinephrine for 48 h (ref. 52).

**In vivo models and echocardiography.** All procedures were carried out in compliance with the standards for the care and use of animal subjects as stated by the requirements of the UK Home Office (ASPA1986 Amendments Regulations 2012) incorporating the EU directive 2010/63/EU. For the in vivo infusion of ISO, male Sprague-Dawley rats (Charles River 310–360 g) were anaesthetized using isoflurane and maintained on a heated pad with monitoring of temperature, pulse oxygenation and ECG (MouseMonitor S, Indus Instruments). Mini-osmotic pumps (model 2001, Alzet, USA) containing ISO (3 mg per kg per day) dissolved in 0.05% ascorbic acid (to prevent the formation of toxic oxidation products) were implanted subcutaneously in the subscapular region, under isoflurane anaesthesia. Control animals were implanted with pumps containing 0.05% ascorbic acid. Rats were infused for 7 days. Echocardiographic indices were obtained according to the recommendations of the British Society of Echocardiography. Transthoracic echocardiography was performed in control and ISO-treated animals with the use of a commercially available Vivid I echocardiography system (GE Healthcare) using an 11.5 MHz phased array 10 S-RS pediatric echo probe. Wall thickness and LV dimensions were obtained from M-Mode measurements taken at the level of the papillary muscles. ISO infusion caused the heart rate of the anaesthetized rats to increase from ∼380 b.p.m. to 510 b.p.m. within 5 min. The respiration rate increased to ∼120 breaths per min and remained elevated for the 7 days. Echo-cardiographic assessment of cardiac function after 7 days of treatment showed that iso-treated rats had a significantly elevated heart rate (499 ± 23 versus 355 ± 26 b.p.m.) and ejection fraction (97 ± 0.8 versus 88 ± 0.9) compared with sham controls. ISO-treated rat hearts were hypertrophied, with a significant

increased thickness of the left ventricular interventricular septum and the posterior wall (Supplementary Table 1).

For the MI procedure, 200–300 g Sprague-Dawley rats were housed individually with access to food ad libitum following surgical procedures. After administration of antibiotics and buprenorphine rats were anaesthetized, before undergoing a thoracotomy and the removal of their pericardium. This allowed the ligation of the left anterior descending coronary artery to induce myocardial infarction (MI). Generally after 16 weeks end-stage heart failure was achieved. To confirm this the ejection fraction was calculated from M-mode echocardiography recordings (Vevo 770 micro-imaging system, Visualsonics). End-stage heart failure was indicated by an ejection fraction of less than 40%. Only animals achieving this level of heart failure were sacrificed to allow isolation of myocytes as described.

No randomization was used. All experiments with cells from in vivo treatments were conducted blind for the treatment.

**Immunofluorescence staining.** Twenty four hours after infection with AKAP18δ-CUTie or TPNI-CUTie adenovirus, ARVM were fixed in paraformaldehyde 4% for 15 min, washed with PBS and permeabilized in 0.2% Triton X-100 for 30 min. After blocking with 1% BSA for 1 h, the cells were incubated with goat anti-SERCA2 (c-20) (sc-8094, Santa Cruz Biotechnology, TX, USA, used at 1:100) and a mouse anti-α-Actinin Sarcomeric (A7811, Sigma-Aldrich, UK, used at 1:2,000), respectively, at 4 °C overnight. Samples were then washed with PBS-0.1% Tween20 and incubated with anti-Goat IgG (H + L) Alexa Fluor 555 conjugate or anti-Mouse IgG (H + L) Alexa Fluor 555 conjugate (A27017 and A-21425 Thermo Fisher Scientific, MA, USA, both used at 1:1,000) for 60 min at room temperature. ARVM expressing AKAP79-CUTie were incubated with Wheat Germ Agglutinin, Alexa Fluor 594 Conjugate (5 µg ml$^{-1}$) (W11262, Thermo Fisher Scientific, MA, USA) for 10 min at 37 °C before fixing the cells with formaldehyde 4%. The fixed cells were rinsed in PBS-0.1% Tween20 and mounted in Ibidi Mounting Medium (Thistle Scientific, UK). Images were acquired with an Inverted Olympus FV1000 confocal microscope.

**Pull-down experiments.** For pull-down experiments ARVM from one heart or $6 \times 10^6$ NRVM were plated onto $3 \times 10$-cm Petri dishes coated with laminin (20 µg ml$^{-1}$) and infected with the indicated adenovirus for 3 h. The cells were washed after 36 h with ADS buffer (106 mM NaCl, 20 mM Hepes, 0.8 mM NaH$_2$PO$_4$, 5.3 mM KCl, 0.4 mM MgSO$_4$, 5 mM glucose) and lysed in RIPA buffer (Sigma-Aldrich, UK) supplemented with Complete EDTA-free protease inhibitor cocktail tablets (Roche Diagnostics Limited, UK) for 5 min on ice. The cells were then collected and placed on a rotating wheel for 20 min at 4 °C. Insoluble material was removed by centrifugation at 9,600g for 10 min at 4 °C and total protein was quantified by Micro BCA Protein Assay Kit (Pierce Biotechnology Inc., IL, USA). Seven hundred fifty milligrams of proteins were rotated for 2–4 h at 4 °C with 30 µl of agarose beads coated with a monoclonal anti-GFP antibody (GFP-Trap_A, gta-10, ChromoTek GmbH, DE). The precipitates were then collected by centrifugation at 16,000g for 3 min and beads were washed five times with ice-cold RIPA buffer. Bound proteins were then eluted in 25 µl of 2 × SDS-loading buffer (Life Technologies) and released from the beads at 95 °C for 5 min. Pulled down proteins were resolved on Bolt 4–12% Bis-Tris Plus Gels (Thermo Fisher Scientific, MA, USA) and transferred onto nitrocellulose membrane (Amersham Protran 0.45 µm, GE Healthcare Life Sciences, UK). After transfer, the membranes were blocked for 1 h at room temperature in Protein-Free (TBS) Blocking buffer (Thermo Fisher Scientific, MA, USA), and then incubated overnight at 4 °C with the following primary antibodies: goat anti-SERCA2 (c-20) (sc-8094, Santa Cruz Biotechnology, TX, USA, used at 1:500); goat anti-Troponin T-C (c-19) (sc-8121, Santa Cruz Biotechnology, TX, USA, used at 1:500) and Adenylyl Cyclase V/VI (C17) (sc-590, Santa Cruz Biotechnology, TX, USA, used at 1:200). After five washes with TBS-0.5% Tween20 (Alfa Aesar, MA, USA) membranes were incubated at room temperature for 2 h with appropriate horseradish peroxidase-conjugated secondary antibodies and detected with ECL western blotting detection kit (Thermo Fisher Scientific, MA, USA). The blots were stripped with stripping buffer (Thermo Fisher Scientific, MA, USA) and reprobed with anti-glyceraldehyde-3-phosphate dehydrogenase (GAPDH) antibody (sc-1666574, Santa Cruz Biotechnology, TX, USA, used at 1:500) to detect protein contamination and with anti-GFP antibody (sc-9996, Santa Cruz Biotechnology, TX, USA, used at 1:1000) to control for efficient pull-down. For all experiments at least three biological replicates were used.

**Western blot analysis.** $2 \times 10^6$ NRVM were plated onto 6-cm Petri dishes coated with laminin (40 µg ml$^{-1}$) and treated as described above. After 48 h cells were washed with a modified Ringer saline solution and then treated at room temperature with ISO 1 nM alone, IBMX 100 µM alone and FRSK 25 µM + IBMX 100 µM as indicated. Cells were lysed using ice-cold RIPA buffer supplemented with Complete EDTA-free protease inhibitor cocktail tablets and PhosSTOP Phosphatase Inhibitor Cocktail Tablet (Roche Diagnostics Limited, UK) on ice and gently agitated for 20 min. Protein concentrations were determined by Micro BCA Protein Assay Kit (Pierce, Rockford, IL). Samples equivalent to 8–30 µg proteins were separated on Bolt 4–12% Bis-Tris Plus Gels after blocking with 5% Phospho Blocker milk (Cell Biolabs Inc., San Diego, CA, USA) for 1 h at room temperature

membranes were incubated overnight at 4 °C with phosphor-specific rabbit anti-TPNI phospho (Ser22 + 23) antibody (1:2,000, ab58545 Abcam, UK) rabbit anti-PLB phospho (Ser16) antibody (1:4,000, A010-12, Badrilla, UK), anti-MyBPC phospho (Ser282) antibody (1:2,500, kindly provided by Dr Sakthivel Sadayappan, Department of Cell and Molecular Physiology, Loyola University Chicago, IL). The blots were washed and incubated with appropriate horseradish peroxidase-conjugated secondary antibodies for 2 h at room temperature. Immunoreactive bands were visualized by ECL western blotting detection kit. Antibody dilutions and exposure times were adjusted to ensure that for each antibody the signal detected was within the linear range. Membranes were then stripped of bound antibodies by incubation with stripping buffer for 45 min at room temperature with gentle agitation. The blots were reprobed with mouse anti-TPNI (1:2,000, ab19615 Abcam, UK), mouse anti-PLB (1:4,000, A010-14 Badrilla, UK), mouse anti-MyBPC3 (1:1,000, sc-137237 Santa Cruz Biotechnology, TX, USA) for loading controls. Quantification of the band intensity was accomplished by densitometry using ImageJ software (http://rsbweb.nih.gov/ij/index.html). The extent of protein phosphorylation was calculated as the ratio of the density of the band corresponding to the phosphorylated protein to the density of the band corresponding to total protein. The values were subsequently normalized using GraphPad Prism software (GraphPad Software, Inc., CA, USA), defining zero as the density after H89 treatment (lowest phospho/total ratio) and 100% as the density at saturation (FRSK 25 µM + IBMX 100 µM treatment, highest phopho/total ratio) for each data set. Densities measured for the other treatments are expressed as a percentage relative to the saturating treatment. Uncropped scans of films are shown in Supplementary Fig. 10.

**Intracellular Ca$^{2+}$ dynamics.** Isolated ARVM were plated onto laminin (40 µg ml$^{-1}$) coated coverslips and cultured overnight. Cells were loaded with 1 µM Fura-2 AM (Molecular Probes, OR, USA) in the dark at room temperature for 15 min and then washed three times. Coverslips were mounted in the perfusion chamber, continuously perfused with a Tyrode solution containing 1.4 mM Ca$^{2+}$ at 37 °C, and paced at 1 Hz using a field stimulator (Myopacer; IonOptix, Milton, MA). When at steady state, cells were perfused with ISO 0.3 nM or IBMX 100 µM for about 10 min. The background-subtracted Fura-2AM emission at 510 nm was recorded and expressed as a ratio of fluorescent light on excitation at 340 and 380 nm (R340/380). The recorded transients were analysed using the IonWizard software (IonOptix, Milton, MA).

**FRET imaging.** FRET imaging experiments were performed 24–48 h after transduction with adenovirus carrying each sensor, as described before[53]. Cells were maintained at room temperature in a modified Ringer solution (125 mM NaCl, 20 mM Hepes, 1 mM Na$_3$PO$_4$, 5 mM KCl, 1 mM MgSO$_4$, 5.5 mM glucose, CaCl$_2$ 1 mM, pH 7.4). ARVM were imaged 18 h after infection and kept at ~35 °C in Tyrode solution containing 1.4 mM Ca$^{2+}$. An inverted microscope (Olympus IX71) with a PlanApoN, 60 ×, NA 1.42 oil immersion objective, 0.17/FN 26.5 (Olympus, UK), was used. The microscope was equipped with a CoolSNAP HQ$^2$ monochrome camera (Photometrics) and a DV2 optical beam-splitter (MAG Biosystems, Photometrics). Images were acquired and processed using MetaFluor 7.1, (Meta Imaging Series, Molecular Devices). FRET changes were measured as changes in the background-subtracted 480 nm/545 nm fluorescence emission intensity on excitation at 430 nm and expressed as $R/R_0$, where $R$ is the ratio at time $t$ and $R_0$ is the average ratio of the first 8 frames. FRET imaging experiments with ARVM isolated from MI rats and age-matched controls were performed using an ORCA-ER CCD camera (Hamamatsu Photonics, Welwyn Garden City, UK) attached to an inverted microscope (Nikon TE2000) equipped with a 30 Watt dia-illuminator. The system has an EX436/20 excitation filter combined with DM455 dichroic mirror.

**Sarcomere shortening.** ARVM with resting sarcomere lengths between 1.50 and 1.80 µm were examined for shortening dynamics. Experiments were performed using an inverted microscope (Olympus IX71) with a PlanApoN, 40 ×, NA 1.30 oil immersion objective. Myocytes were imaged by using the transillumination light path equipped with an additional red filter and recorded by a MyoCam-S CCD video camera (IonOptix Milton, MA). Cells were electrically stimulated at 1 Hz in the presence of 1.4 mM Ca$^{2+}$ at 35 ± 1 °C. Amplitude and velocity of sarcomere shortening/relengthening was measured and analysed using the video-based sarcomere length detection software module of IonWizard (IonOptix Milton, MA). Parameters from 10 contractions were averaged to minimize beat-to-beat variation and obtain mean values at baseline (control) and for the response to stimulus.

A separate cohort of myocytes were infected with Ad-EPAC-S$^{H187}$ (ref. 22) and used to measure simultaneously sarcomere shortening and FRET. Cells plated onto laminin-coated coverslips were mounted in a metal chamber and kept in Tyrode solution containing 1.4 mM Ca$^{2+}$ at 35 ± 1 °C. FRET and shortening images were separated by an additional beam splitter (600 nm HP dichroic mirror, Cairn, Kent, UK) in the emission path and acquired simultaneously by a CoolSNAP HQ$^2$ and MyoCam, respectively. The two cameras were connected to separate PCs to run the acquisition in parallel. FRET was performed as described above. In our experiments the simultaneous acquisition of shortening data did not affect the quality of the FRET acquisitions. After FRET and contractility signals reached the steady state

(after about 2 min), ISO 0.3 nM or IBMX 100 μM were directly applied into the bath solution. Sarcomere shortening amplitude values before and after the stimuli were used to determine the percentage increase of stimulated shortening.

**Mathematical modelling and simulation.** We developed a mouse ECC *in silico* model by embedding the Negroni *et al.* five-state contraction model[23] within the well-established Morotti–Grandi[24] mouse ventricular action potential, $Ca^{2+}$ and β-adrenergic signalling model. In the Supplementary Notes, we provide more details on the model formulation describing the merging of the models and emphasizing that this new model recapitulates measured changes in AP, $[Ca^{2+}]$ and $[Na^+]$ in response to ISO. Supplementary Fig. 7a shows the model schematic and rodent-specific PKA targets, which includes: LTCC, slowly inactivating $K^+$ current, ryanodine receptors, phospholemman, PLB, Inhibitor-1 and myofilament proteins (TPNI, MyBPC, Titin). All simulations were performed in MATLAB (The MathWorks, Natick, MA, USA) using the stiff ordinary differential equation solver ode15 s. All source codes used in the paper are available online at https://somapp.ucdmc.ucdavis.edu/Pharmacology/bers/.

**Statistical analysis.** Data are presented as means ± s.e.m. Sample size was chosen based on previous data to ensure adequate power to detect 10% difference. Statistical analysis was performed with GraphPad Prism 5.0. The number of technical and biological replicates is indicated in the figure legends. All groups that were statistically compared showed equal variance. One-way ANOVAs with *post hoc* test or unpaired two-tailed Student's *t*-tests were used as appropriate. Statistical significance, when achieved, is indicated as $*P \le 0.05$, $**P \le 0.01$, $***P \le 0.001$.

**Data availability.** The data that support the findings of this study are available from the corresponding author upon reasonable request. Source codes for the entire mathematical model used in the paper are freely available online at: https://somapp.ucdmc.ucdavis.edu/Pharmacology/bers/.

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

## Acknowledgements

This work was supported by the Fondation Leducq (O6 CVD 02), the British Heart Foundation (BHF; PG/10/75/28537, RG/12/3/29423 and PG/15/5/31110) and BHF Centre of Research Excellence, Oxford (RE/08/004) to M.Z.; FOCEM (MERCOSUR Structural Convergence Fund), COF 03/11 and the National Scientific Program of ANII (SNI) to M.R.M. and S.P.; the Conselho Nacional de Desenvolvimento Científico e Tecnológico (CNPq) Scholarship to M.B.; British Heart Foundation (JG-12/18/30088 and RM/13/1/30157) to J.G. and Medical Research Council (J.G., P.W. MR/L006855/1); A.H.A. American Heart Association (15SDG24910015 to E.G. and 2014POST18380011 to S.M.); NIH grants R01-HL105242, HL030077 (D.M.B.) and HL131517 (E.G.). The authors would like to thank Kees Jalink, The Netherlands Cancer Institute, Amsterdam, for providing the EPAC-S[H187] construct, Miles Houslay, Kings College, London for useful discussion, and Lia Margiotta for help with the artwork.

## Author contributions

M.Z., S.P., D.M.B., N.C.S., Ma.Be., A.K. and E.G. designed experiments. N.C.S., D.M.B., A.K., M.Be., M.Br., M.R.M. and S.M. performed experiments and analysed the data. M.Z. and S.P. wrote the manuscript. C.C., P.W. and J.G. provided *in vivo* rat models of cardiac disease. M.Z. conceived research question and oversaw entirety of research.

## Additional information

**Competing interests:** The authors declare no competing financial interests.

