## [Peer Review File · Nature Communications]

Reviewers' comments:

Reviewer #1 (Remarks to the Author):

In their elegant study, Nicoletta Surdo and coworkers present a molecular design of a universal FRET tag which can be used to target available cytosolic biosensors to specific subcellular microdomains (or nanodomains) without changing their dynamic range and sensitivity. Using this strategy, concentrations and real-time dynamics of second messengers, as exemplified by cAMP measurements by CUTie sensors in this study, can be directly compared between various subcellular locales which is an important advantage to study compartmentalized signaling. Using such sensors, the authors uncovered that the same receptor stimulus (beta-adrenergic receptor activation by isoproterenol) can trigger distinct cAMP responses in various microdomains. For example, more cAMP is generated at the membrane or at SERCA and less cAMP is detected at TnI than in bulk cytosol. Furthermore, this differential regulation of cAMP signals has been found necessary for optimal stimulation of cardiomyocyte contractility. Interestingly, beta-adrenergic desensitization caused by catecholamine treatment in an in vitro cardiomyocyte hypertrophy model differentially affect sarcomeric vs. sarcolemmal or sarcoplasmic which provides new intriguing insights into mechanisms of disease and potential therapeutic strategies. This story has a potential to significantly influence thinking in the field of cardiology. The molecular tools and major findings of the paper are novel and they will be of high interest to other researchers in the field. To strengthen the revised manuscript in terms of conclusions and data presentation, the authors should refer to the concerns listed below.

Statistical tests reported in the Figures seem appropriate. Error bars and probability values are accurately described in all figure legends.

Major points:

1. There is a big discrepancy between the dynamic range of the new sensors measured in cAMP microinjected cells and in intact cells stimulated with FRSK+IBMX, see Fig. 1d/g - 12% FRET change in cells - and Fig 1e 20% in microinfusion experiments. This dramatic difference should be much better explained. Together with the $EC_{50}=7.4 \mu\text{M}$ for cAMP, one gets an impression that the sensor has a very low affinity and is not saturated in cells even by FRSK/IBMX. However, the situation is much better in ARVM in Fig 3 or in NRVM - Sfig3. The FRSK/IBMX responses in Fig 1g are quite slow - 8 min to reach max response as compared to 5 min in Fig 1b (Epac1-camps sensors). The issue of sensitivity culminates when another older EPAC-S^{H187} sensor is being ironically used in Fig 4a to detect cAMP response to low ISO concentration, not detectable by CUTie. I would recommend to discuss the sensitivity issue much more detailed and carefully and provide concentration-response curve for ISO-stimulated ARVMs expressing cytosolic and targeted CUTie versions. Comparing kinetics of different sensor responses at subsaturating concentrations are sometimes difficult too, it would be helpful to show similar experiments under saturating ISO as well.
2. In figure 2, IF images of ARVMs and IPs plus FRET data are shown for NRVMs. Since adult cells are most relevant to compartmentation studies in term of microdomains structural organization etc, proper targeting of the sensors should be ideally confirmed in adult cells, e.g. IPs done by using GFP-trapped samples from adenovirus-transduced ARVMs. Since some IF images in Fig 2b show sensor aggregates, proper localization proof in adult cells is important.
3. Fig. 6. Based on the same considerations as above (the maturity of microdomains organization), in vivo adult disease model data in addition to in vitro NRVM hypertrophy results should be provided. Any relevant in vivo disease model model such as hypertrophied ARVMs or AMVMs from animals after catecholamine minipumps implantation, TAC or MI more would be extremely helpful.

Minor points:

1. In this reviewer's opinion, the word "nano-domains" used by authors could be replaced by "microdomains". The value of 500 nm mentioned by the authors (which is in low micrometer range) is closer to 1 μm than to 1 nm, too big to call them nano-domains, at least in my

perception. The phrase "nanoscopic heterogeneity" in the abstract sound a bit odd.

2. In Fig. 1b (Page4, line 11), when describing targeted Epac1-camps constructs, no detectable response could be seen with PDE4A1-Epac1-camps, giving an example of a sensor which did not work for targeting using a conventional strategy. A similar construct has been published before by Herget S et al. Cell Signal 2008, PMID 18467075, which did not show any FRET change, unless the sequence of PDE4A1 and Epac1-camps was switch to Epac1-camps- PDE4A1. All other constructs had a dynamic range reduced by half similarly to what has been also reported by the same group for RI- and RII-epac sensors - Di Benedetto et al. Circ Res 2008, PMID: 18757829. Maybe this paper can be also cited and briefly mentioned in context or comparing signals in different compartments (in addition to Ref. 13). To truly compare the kinetics of different sensors in Fig 1b, I would have normalized all traces from 0 to 100% before saying that they are very different.

3. Page5 line 24. The text states that PKA phosphorylation of LTCC and PLB causes larger amplitudes of calcium transient and contraction. I thought PLB phosphorylation changes mostly the time of decay, not so much the amplitude

4. 2. Please, check BrE/AmE spelling and use AmE throughout the manuscript. The current version contains a mixture of such words as for example compartmentalised, compartmentalization, localized, localisation etc. Some colons are missing per AmE style, e.g. "In the heart,..." Change also "Plasmamembrane" to "plasma membrane" or "plasmalemma"

Reviewer #2 (Remarks to the Author):

This study describes an elegant approach to design novel targeted cAMP FRET probes (CUTie) which in principle should have similar dynamics and kinetics in response to a similar rise in local [cAMP]. The author used three CUTie probes, two targeted respectively to sarcolemma and SR membranes, and one targeted to troponin I (TPNI), expressed in adult cardiomyocytes and compared the kinetics and amplitudes of the localized cAMP signals in response to isoproterenol (ISO) or to IBMX (a non-selective PDE inhibitor). They conclude that [cAMP] in the TPNI compartment is lower than in the two other compartments upon ISO application and conclude that this is necessary to optimize cardiac contractility upon adrenergic activation. The experiments are elegantly done, but for the reasons detailed below, the interpretation of the results appears too speculative.

Specific:

Fig. 1: This figure shows clearly that the CUTie constructs with various targeting domains show similar dynamics and kinetics while the Epac1-camps constructs with targeting domains vary significantly from one construct to another. However, the experiments were performed in CHO cells where I assume that most of the target proteins on which the constructs are supposed to bind are absent. The data show therefore that the CUTie constructs respond similarly when expressed in the cytosol but do not demonstrate that the dynamics and kinetics would be identical upon a sudden rise in cAMP when the constructs are immobilized on their targets.

Fig. 2f: In the same vein, this figure shows that the in-cell cAMP concentration-response curves for three CUTie constructs are superimposable in CHO cells, not when the probes are immobilized on their targets in cardiomyocytes. While Fig. 2e indicates that the response to 1 mM cAMP and 100 μ M IBMX in the patch pipette produced a similar response in ARVMs, it would be important to show that the concentration-response curves for the FRET changes in response to cAMP are identical for the three CUTie probes when expressed in ARVMs.

Supplemental Fig. 4a: the lack of response of the three targeted CUTie probes to 0.3 nM ISO (while the cytosolic untargeted FRET reporter EPAC-SH187 shows a clear response) is odd, considering that Fig. 4a, c and e show that this concentration of ISO is sufficient to produce a maximal response on sarcomere shortening and Ca²⁺ transients. It is therefore speculative to

conclude that the local [cAMP] at TPNI is lower than at SR or sarcolemmal membranes. According to Fig. 2f, the EC50 of the three CUTie probes for cAMP is around 7 μ M. One may wonder then whether these targeted probes are sensitive enough to detect physiological changes in cAMP concentration?

Fig. 4c and d: The experiment shown in (c) shows a 20-25% reduced response to IBMX vs. ISO, which is not representative of the summary data shown in (e) which shows a 60% average reduced response to IBMX.

Fig. 6: This set of experiments was performed in neonatal cells, while the rest of the data presented in the main manuscript was obtained in adult ventricular cells. Why? While the conclusions of these experiments comfort the authors hypothesis of a lower [cAMP] at TPNI as compared to sarcolemmal and SR membranes, the concentration and distribution of cAMP in response to ISO is clearly different in neonatal and adult cells. For instance, 0.5 nM ISO produces a clear response at all CUTie probes in neonatal cells but 0.3 nM ISO produces undetectable changes in [cAMP] in adult cells. It is therefore difficult to draw conclusions on the changes induced by cellular hypertrophy in this model.

General: Unlike what is shown in Fig. 3d here, Li et al. (Am J Physiol Heart Circ Physiol. 2000;278:H769-H779) showed a similar kinetic in TPNI and PLB phosphorylation upon application of ISO. Moreover, the contribution of TPNI phosphorylation to relaxation has been shown to depend strongly on mechanical load (see also Layland et al., Cardiovasc Res. 2005;66:12-21). Since all the experiments reported in this study were performed in unloaded cells, some caution is required in the interpretation of the results.

Reviewer #3 (Remarks to the Author):

This manuscript introduces an interesting biosensor design that improves the study of cAMP microdomains by reducing the effects of protein fusions on FRET responses. By inserting the Fluorescent protein into the center of the biosensor, this biosensor does not exhibit changes in the dynamic range when tethered to different microdomains as seen previously. This targetable biosensor with improved fidelity was then used to examine cAMP microdomains within cardiomyocytes. By targeting this CUTie biosensor to AKAP79, AKAP18 δ , and TPNI, the authors demonstrate that the TPNI microdomain exhibits a decreased cAMP accumulation in response to 5 nM ISO stimulation. It was observed that 0.3 nM of ISO and 100 μ M IBMX in ARVM both elicit a measurable cytosolic cAMP response using a different biosensor, EPAC-SH187. Within these same ARVMs the contractile shortening was measured and the IBMX stimulation was observed to exhibit a decreased sarcomere shortening compared to the 0.3 nM ISO dose. This lead the authors to hypothesize that cardiomyocytes have optimized contractile ejection volume by increasing the PKA phosphorylation at targets that increase contractile strength while simultaneously allowing the TPNI microdomain to have low phosphorylation to promote increased contractile relaxation. To address this hypothesis, the authors combined two previously developed computational models of ECC and probed the effects of differential cAMP compartmental regulation on ECC. Interestingly, this model suggested that phosphorylation of other PKA targets (PLB, MyBPC3) need to be high but TPNI phosphorylation needs to be low to maximize contractile response to ISO. This model derived hypothesis was then validated with western blots. These data led the authors to test the effectiveness of PDE inhibitors on treating heart failure. Indeed, negative effects of PDE inhibition on TPNI hypophosphorylation suggest that PDE inhibition may not be an appropriate treatment for heart failure.

This paper presents a novel biosensor design that improves the fidelity of targeted cAMP biosensors which may serve as a template for improving other biosensors. However, the conclusions on cardiac maximal contractility are not supported by the experimental data, the stated impact of the biosensor on spatial resolution is misleading and no model details were

provided with the paper. These concerns are detailed below:

Major Concerns:

1. Conclusions of maximal contractility arising from cAMP microdomain differences are not supported by the current experimental data. The reduced FRET ratio change for TPNI-CUTie compared to AKAP18 δ - and AKAP79-CUTie was observed in response to 5 nM ISO (Figure 3) but the differences in sarcomere shortening were observed using 0.3 nM ISO (Figure 4). Also, the 0.5 nM ISO dose used on NRVMs in Figure 6 appears to have a decreased response from TPNI-CUTie but the statistical significance is not tested. In order to begin to correlate the CUTie response differences with changes in fractional shortening the same ISO dose needs to be used between the two experiments. Furthermore, these two experiments are correlative and do not directly test the hypothesis that the TPNI cAMP difference leads to a maximal enhancement of contraction and relaxation. To directly test this hypothesis, the authors could try to disrupt this TPNI compartmentalization (possibly by identifying the PDE regulating this compartment and inhibiting that PDE isoform). The computational model does provide some evidence that the cAMP compartmentalization at TnI is important for maximal sarcomere shortening but the experimental evidence does not directly test this hypothesis.

2. "The CUTie biosensor increases the spatial resolution cAMP sensing" is an overstatement and misleading. This biosensor improves the fidelity of the cAMP sensor when fused with other proteins but the resolution remains the same as previously developed probes. The introduction and discussion both make numerous references to resolution improvements but the only reference to "resolution" in the results comes in the first sentence. The improvement in fidelity though this type of biosensor design is interesting enough that overstating the spatial resolution detracts from the paper. This paper needs to be rewritten with more precise and accurate descriptions of the impact of the CUTie biosensor.

3. Computational model details are not included in the supplement of the paper. It is very important that the computational model details are published with the paper. While the models referenced have been previously published, this paper states that the model used involved merging two models, thus creating a unique model which must be fully explained. The validity of any model assumptions made when merging and modifying the previous models cannot be evaluated in this review as no details were provided.

Additional Points:

- The authors hypothesize that cAMP compartmentalization in the TPNI is PDE driven. This hypothesis can be tested by synthetically compartmentalizing PDE with the CUTie biosensor, which they have already developed with their PDE4A1-CUTie construct. Thus, the Iso response of PDE4A1-CUTie should be compared to untargeted CUTie.
- The decision to use 0.3 nM ISO was based on EPAC-SH187. What is the untargeted CUTie biosensor response to 0.3 nM ISO?
- What is meant by "global cAMP response" (pg 5 ln 32)?
- Is it possible to validate the model predictions with the CUTie biosensor as well (i.e. MyBPC-CUTie)?

A detailed point-by-point response to the reviewers' comments follows below. In our response we have highlighted in bold Figures and tables containing new data as well as the position where we made changes to the text.

Point-by-point response to the referees' comments:

Reviewer #1:

We would like to thank this reviewer for praising our study as having the 'potential to significantly influence thinking in the field of cardiology'.

Major points:

1. *There is a big discrepancy between the dynamic range of the new sensors measured in cAMP microinjected cells and in intact cells stimulated with FRSK+IBMX, see Fig. 1d/g - 12% FRET change in cells - and Fig 1e 20% in microinfusion experiments. This dramatic difference should be much better explained. Together with the $EC_{50}=7.4 \mu M$ for cAMP, one gets an impression that the sensor has a very low affinity and is not saturated in cells even by FRSK/IBMX.*

We thank this reviewer for bringing to our attention this inconsistency in the data. The discrepancy is due to the fact that the data originally presented in Fig. 1d/g were acquired using an imaging system different from that used for the microinfusion data shown in Fig 1e. FRET change in CHO cells on application of FRSK/IBMX (Fig 1d/g) was acquired early on in the characterisation of the CUTie sensor with a system using a mercury lamp as the source of excitation light (as opposed to LED in our current system) and slightly different filter sets, resulting in different values for emission fluorescence intensities and therefore different ratio values. We agree that such discrepancy may be confusing to the reader and we have now repeated the FRSK/IBMX saturation experiments in CHO cells using the same system that we used to generate the curve in Fig 1e (and the other data in this manuscript). The values presented in the **new Fig 1d/g** show no significant difference when compared to the values shown in Fig 1e.

However, the situation is much better in ARVM in Fig 3 or in NRVM - Sfig3.

Indeed these experiments were all performed using our more recent imaging setup.

The FRSK/IBMX responses in Fig 1g are quite slow - 8 min to reach max response as compared to 5 min in Fig 1b (Epac1-camps sensors).

When comparing Fig 1b with the original Fig 1g one may indeed have the impression that CUTie fusions are significantly slower than Epac1-camps fusions. The difference in kinetics is however only apparent and is due to the way the data were presented in the two graphs with respect to time of stimulus application. When the curves are normalised for time of stimulus application (as in Fig 1b and the **new Fig 1g**) this difference disappears. Similar kinetics of FRET change for the two sensors are also confirmed by calculating the $t_{1/2}$ to maximal response which shows no significant difference ($p=0.09$) between Epac1-camps targeted sensors ($t_{1/2} = 71.6 \pm 5.3$) and CUTie targeted sensors ($t_{1/2} = 84.6 \pm 4.3$)

The issue of sensitivity culminates when another older EPAC-S^H187 sensor is being ironically used in Fig 4a to detect cAMP response to low ISO concentration, not detectable by CUTie. I would recommend to discuss the sensitivity issue much more detailed and carefully and provide concentration-response curve for ISO-stimulated ARVMs expressing cytosolic and targeted CUTie versions. Comparing kinetics of different sensor responses at subsaturating concentrations are sometimes difficult too, it would be helpful to show similar experiments under saturating ISO as well.

In response to this reviewer's concern regarding the sensitivity of CUTie we remark that, although

CUTie EC₅₀ for cAMP is higher compared to some of the other available cAMP FRET-based sensors, it is sufficient to reliably detect cAMP changes elicited by 0.5 nM ISO (summarised in Tab 1), a concentration of agonist that is well in the physiological range. A comment specifically addressing the sensitivity of CUTie has been included in the discussion (**p12, I6**). In addition, we provide in the **new Suppl Fig 3** a concentration-response analysis for the different sensors at increasing concentrations of ISO. These new data show that, even at saturating ISO (1 μ M – 100 μ M range) the cAMP change detected in the bulk cytosol and at TPNI is significantly lower than at AKAP79 and AKAP18 δ .

2. In figure 2, IF images of ARVMs and IPs plus FRET data are shown for NRVMs. Since adult cells are most relevant to compartmentation studies in term of microdomains structural organization etc, proper targeting of the sensors should be ideally confirmed in adult cells, e.g. IPs done by using GFP-trapped samples from adenovirus-transduced ARVMs. Since some IF images in Fig 2b show sensor aggregates, proper localization proof in adult cells is important.

Our FRET data in NRVM suggest that the local regulation of cAMP signal in response to ISO is independent of the highly structured and organised microanatomy of ARVM, as we find identical results in the less organised NRVM as in ARVM. We believe the intracellular localisation signal observed in cell expressing AKAP79-CUTie is not due to aggregation of the sensor but largely to genuine localisation to the T-tubular system. However, to demonstrate appropriate targeting of the sensors in adult myocytes we have now performed co-IPs from ARVM expressing the sensors and confirmed the expected localisation. These data are now presented in the **new Fig 2c**, whereas the co-IPs from NRVM are now shown in Suppl Fig2.

3. Fig. 6. Based on the same considerations as above (the maturity of microdomains organization), in vivo adult disease model data in addition to in vitro NRVM hypertrophy results should be provided. Any relevant in vivo disease model such as hypertrophied ARVMs or AMVMs from animals after catecholamine minipumps implantation, TAC or MI more would be extremely helpful.

The reviewer raises here an important point concerning possible differences in the organisation of cAMP local domains in different models of disease. As suggested, we have now performed FRET imaging experiments using myocytes from rats subjected to ISO minipump infusion as well as myocytes from a rat MI model. The new data are shown in **Fig 6d-g**. The new experiments confirm our findings with the *in vitro* hypertrophy model and show a dramatic reduction in the level of cAMP at TPNI. The new data also demonstrate that, in contrast to the *in vitro* model of hypertrophy, both *in vivo* models show a significant reduction of cAMP at AKAP18 δ in diseased vs healthy myocytes, although this is not as large as the reduction observed at the myofilaments. These findings provide additional original insight into mechanisms of disease and point to different functional effects on local signalling that may derive from β -adrenergic desensitization caused by short-term catecholamine exposure versus myocardium remodelling events that characterise chronic stress and the process leading to heart failure. These new findings and their relevance is now discussed **on p11, I5 and p14, I16**.

Minor points:

1. In this reviewer's opinion, the word "nano-domains" used by authors could be replaced by "microdomains". The value of 500 nm mentioned by the authors (which is in low micrometer range) is closer to 1 μ m than to 1 nm, too big to call them nano-domains, at least in my perception. The phrase "nanoscopic heterogeneity" in the abstract sound a bit odd.

With the term nano-domain we intended to convey the notion that signalling by cAMP really happens in a very limited space surrounding relevant targets of PKA. The upper bound of 300nm that we mention in the paper is very likely an overestimation of the real size of these domains. This is based on the consideration that the distance between myofibrils and SR is in the order of tens of

nanometres and yet the signal detected by AKAP18 δ -CUTie is larger than the signal detected at TPNI. On the other hand, reticular SR and T tubule can be over 1 μ m apart, and yet AKAP18 δ -CUTie and AKAP79-CUTie detect the same cAMP change. These considerations suggest that the differences we observe are not dependent on the physical distance between the various sites but are the result of very tight local regulation within a very narrow volume surrounding the individual multiprotein complexes targeted by CUTie. To convey our point more clearly, we now discuss this point more in detail on **p12, I21**.

2. In Fig. 1b (Page4, line 11), when describing targeted *Epac1-camps* constructs, no detectable response could be seen with *PDE4A1-Epac1-camps*, giving an example of a sensor which did not work for targeting using a conventional strategy. A similar construct has been published before by Herget S et al. *Cell Signal* 2008, PMID 18467075, which did not show any FRET change, unless the sequence of *PDE4A1* and *Epac1-camps* was switch to *Epac1-camps- PDE4A1*. All other constructs had a dynamic range reduced by half similarly to what has been also reported by the same group for RI- and RII-*epac* sensors - Di Benedetto et al. *Circ Res* 2008, PMID: 18757829. Maybe this paper can be also cited and briefly mentioned in context or comparing signals in different compartments (in addition to Ref. 13).

This reference has now been included

To truly compare the kinetics of different sensors in Fig 1b, I would have normalized all traces from 0 to 100% before saying that they are very different.

After normalisation the difference in kinetics resulted to be not significant. Therefore we have deleted from the reference to a difference in kinetics of response for the *Epac1-camps* targeted reporters the main text.

3. Page5 line 24. The text states that PKA phosphorylation of LTCC and PLB causes larger amplitudes of calcium transient and contraction. I thought PLB phosphorylation changes mostly the time of decay, not so much the amplitude

It is true that PLB phosphorylation dramatically accelerates SR Ca uptake, $[Ca]_i$ decline and relaxation (the main cause of the lusitropic PKA effect). However, the stronger SR Ca uptake also increases SR Ca content, and thus the SR Ca available for release. That would increase Ca transient amplitude and contraction even if the same fractional SR Ca release were to occur. But because fractional SR Ca release is enhanced by both higher SR Ca content and Ca current trigger, the Ca transient amplitude and inotropic effect is further enhanced (Bassani et al., *Am J Physiol*, 268:C1313-9, 1995; Shannon et al. *Biophys J*. 78:334-43, 2000). Note that the faster SERCA function during $[Ca]_i$ decline competes more effectively with the Na/Ca exchange (the main competitor for Ca removal), allowing a larger fraction of Ca to be taken back into the SR as well. That also helps the SR accumulate some of the higher Ca influx via LTCC. Direct demonstrations of these effects are in a Kranias-Bers paper (*Circ Res* 92:769-76, 2003) using mice expressing only PLB that cannot be phosphorylated by PKA or CaMKII (PLB- ST16/17AA mice). The ISO effect on $t_{1/2}$ of $[Ca]_i$ decline (and lusitropy) was abolished, and the ISO effect to enhance Ca transient amplitude and contraction was reduced by ~50%.

4. 2. Please, check BrE/AmE spelling and use AmE throughout the manuscript. The current version contains a mixture of such words as for example compartmentalised, compartmentalization, localized, localisation etc. Some colons are missing per AmE style, e.g. "In the heart,..." Change also "Plasmamembrane" to "plasma membrane" or "plasmalemma"

The text has now been carefully checked for consistency and typos.

Reviewer #2

Fig. 1: This figure shows clearly that the CUTie constructs with various targeting domains show similar dynamics and kinetics while the Epac1-camps constructs with targeting domains vary significantly from one construct to another. However, the experiments were performed in CHO cells where I assume that most of the target proteins on which the constructs are supposed to bind are absent. The data show therefore that the CUTie constructs respond similarly when expressed in the cytosol but do not demonstrate that the dynamics and kinetics would be identical upon a sudden rise in cAMP when the constructs are immobilized on their targets.

Fig. 2f: In the same vein, this figure shows that the in-cell cAMP concentration-response curves for three CUTie constructs are superimposable in CHO cells, not when the probes are immobilized on their targets in cardiomyocytes. While Fig. 2e indicates that the response to 1 mM cAMP and 100 μ M IBMX in the patch pipette produced a similar response in ARVMs, it would be important to show that the concentration-response curves for the FRET changes in response to cAMP are identical for the three CUTie probes when expressed in ARVMs.

This reviewer raises here a very valid point. There is however a good reason why we performed the calibration curves in CHO cells. We found that it was impossible to perform the calibration experiments in ARVM as the cells hypercontract and die shortly after establishing the whole cell configuration (panel c in figure 1 below). This effect was particularly obvious when cAMP concentrations of 30 μ M and above were microinfused. In addition, for lower concentrations of cAMP, it was typically difficult to equilibrate the cell uniformly with cAMP, probably due to the complex subcellular organization of adult cardiomyocytes.

Figure 1: a) Differential interference contrast image of a patch-clamped adult rat ventricular myocyte. The patch-clamp pipette can be seen coming from the left. b) FRET-ratio image of the same cell before establishing the direct access to the cytosol, directly after (c) and at 200 and 450 seconds after (d, e). In addition the sequence shows that immediately after the establishment of the whole-cell configuration the cell shrinks until after some minutes it finally dies.

NRVM tolerated the microinfusion with up to 1 mM cAMP without visible damage. However with these cells we encountered a different problem. For the calibration experiments it is necessary to inhibit the PDEs to avoid rapid degradation of the microinfused cAMP, particularly at lower [cAMP]. We found that application of 100 μ M IBMX by itself results in a FRET change of about 10% (similar to the FRET change generated by microinfusion of 10 μ M cAMP, see **new supplementary Fig 2c**). This is due to the high basal activity of adenylyl cyclases (ACs) and PDEs in these cells. For this reason, it was impossible to obtain FRET change values for [cAMP] below 10 μ M. Therefore, to obtain a full calibration curve, we opted for CHO cells which show minimal basal activity of ACs and PDEs. However, as it was still possible to reliably measure in NRVM FRET changes for cAMP concentration above 10 μ M, in the **new supplementary Fig 2c** we address the point raised by this reviewer by showing that, in the presence of IBMX (when compartmentalisation is disrupted), the three CUTie sensors generate an identical FRET change (both in terms of amplitude and kinetics) even when they are immobilised on their targets.

Supplemental Fig. 4a: the lack of response of the three targeted CUTie probes to 0.3 nM ISO (while the cytosolic untargeted FRET reporter EPAC-SH187 shows a clear response) is odd, considering that Fig. 4a, c and e show that this concentration of ISO is sufficient to produce a maximal response on sarcomere shortening and Ca^{2+} transients. It is therefore speculative to conclude that the local [cAMP] at TPNI is lower than at SR or sarcolemmal membranes. According to Fig. 2f, the EC50 of the three CUTie probes for cAMP is around 7 μ M. One may wonder then whether these targeted probes are sensitive enough to detect physiological changes in cAMP concentration?

While it is true that the sensitivity of CUTie is not high enough to detect the local increase in [cAMP] generated by application of 0.3nM ISO, this sensor can reliably measure the increase in cAMP in response to 0.5nM ISO, a concentration that is over three orders of magnitude lower than the saturating concentration ($\geq 1\mu$ M, see **new Suppl Fig 3**). In addition, it should be noted that at 0.3nM ISO we do not observe maximal increase in contractility (compare Fig 4c,d with the **new Suppl Fig 5c, d**). The reason to show the contractility data at 0.3 nM is that at this concentration ISO does not generate a significantly larger overall increase in cAMP compared to IBMX, which may then account for the larger effect on contractility. With 0.3 nM ISO we see a larger increase in contractility compared to 100 μ M IBMX even if ISO at this concentration generates in the bulk cytosol less cAMP than IBMX. However, we have now also performed experiment using 5nM ISO (a concentration that generates an increase in cAMP that we can clearly detect with the targeted sensors). These new data are presented in **Suppl Fig 5** and confirm that a compartmentalised increase in cAMP is significantly more efficient in increasing contractility than the homogeneous increase in cAMP generated by 100 μ M IBMX. For the sensitivity of CUTie please see the discussion on this issue in the response to Reviewer #1

Fig. 4c and d: The experiment shown in (c) shows a 20-25% reduced response to IBMX vs. ISO, which is not representative of the summary data shown in (e) which shows a 60% average reduced response to IBMX.

There is a misunderstanding here. Panel c in Fig 4 does not show representative curves for the data summarised in panel d but both panels show mean values expressed in a different way. Panel c shows normalised (to the value before contraction) mean sarcomere shortening kinetics measured at steady state after the application of ISO or IBMX. Panel d shows sarcomere shortening as percent increment over control (before application of the stimulus). Values are calculated as $(\Delta \text{shortening} / \text{shortening}_{\text{before stimulus}}) * 100$, where $\Delta \text{shortening} = (\text{shortening}_{\text{stimulated}} - \text{shortening}_{\text{before stimulus}})$.

Fig. 6: This set of experiments was performed in neonatal cells, while the rest of the data presented in the main manuscript was obtained in adult ventricular cells. Why?

We choose to use NRVM here because the hypertrophy induced in vitro is more robust when NE is applied to neonatal rather than adult cell (see for example Zoccarato et al Circ Res 2016).

While the conclusions of these experiments comfort the authors hypothesis of a lower [cAMP] at TPNI as compared to sarcolemmal and SR membranes, the concentration and distribution of cAMP in response to ISO is clearly different in neonatal and adult cells. For instance, 0.5 nM ISO produces a clear response at all CUTie probes in neonatal cells but 0.3 nM ISO produces undetectable changes in [cAMP] in adult cells. It is therefore difficult to draw conclusions on the changes induced by cellular hypertrophy in this model.

In further support of our conclusions we now provide data generated using two in vivo disease models. For the new set of data presented in the **new Fig 6d,e**, ARVM from animals subjected to MI or ISO minipump infusion were examined by FRET imaging and compared with healthy ARVM controls. These new data confirm a significantly lower cAMP response at TPNI compared to the

other two compartments in diseased vs healthy myocytes.

General: Unlike what is shown in Fig. 3d here, Li et al. (Am J Physiol Heart Circ Physiol. 2000;278:H769-H779) showed a similar kinetic in TPNI and PLB phosphorylation upon application of ISO.

Part of the discrepancy is that Li et al. used 1 μ M ISO, to intentionally obtain very rapid and maximal PKA-dependent phosphorylation of both PLB and TPNI for their purposes (one of us was involved). Here we use 1000x lower concentration (1 nM), that is closer to a physiological range and is more likely to detect submaximal or slower effects at different target sites. Their 32 P assay (vs. our target site specific antibodies) could also include phosphorylation by kinases other than PKA, that could be activated by that strong ISO treatment (e.g. CaMKII, PKC, PAK3).

Moreover, the contribution of TPNI phosphorylation to relaxation has been shown to depend strongly on mechanical load (see also Layland et al., Cardiovasc Res. 2005;66:12-21). Since all the experiments reported in this study were performed in unloaded cells, some caution is required in the interpretation of the results.

The reviewer and the Layland et al. review are correct. Indeed, the same Bers-Kranias study mentioned above (Li et al. 2000, Am J Physiol, 278:H769-79) showed that PLB phosphorylation was required for detectable β -AR-induced lusitropy in unloaded myocytes, but that during isometric contractions TPNI could contribute significantly to lusitropy. In the revised manuscript we have clarified this, adding a cautionary note (**p13, I23**) that unloaded myocyte shortening is likely to underestimate the *in vivo* effect of TPNI phosphorylation on relaxation kinetics. We would like eventually to look at this in follow-up studies in the presence of afterload, but that is beyond our present scope here. These new optical tools help to make that feasible.

Reviewer #3 (Remarks to the Author):

Major Concerns:

1. Conclusions of maximal contractility arising from cAMP microdomain differences are not supported by the current experimental data. The reduced FRET ratio change for TPNI-CUTie compared to AKAP18 δ - and AKAP79-CUTie was observed in response to 5 nM ISO (Figure 3) but the differences in sarcomere shortening were observed using 0.3 nM ISO (Figure 4). Also, the 0.5 nM ISO dose used on NRVMs in Figure 6 appears to have a decreased response from TPNI-CUTie but the statistical significance is not tested. In order to begin to correlate the CUTie response differences with changes in fractional shortening the same ISO dose needs to be used between the two experiments.

We do feel that our results are internally consistent and are supported by the experimental data. We have made numerous clarifications and additional experiments to clarify points like this. To address this point in particular, we now present a new set of data in **Suppl Fig 5** where we measure contractility on application of 5 nM ISO or 100 μ M IBMX (same concentrations used in Figure 3). The new set of data show that application of 5nM ISO generates an even larger increase in contractility (when compared to 0.3nM ISO) than 100 μ M IBMX.

Furthermore, these two experiments are correlative and do not directly test the hypothesis that the TPNI cAMP difference leads to a maximal enhancement of contraction and relaxation. To directly test this hypothesis, the authors could try to disrupt this TPNI compartmentalization (possibly by identifying the PDE regulating this compartment and inhibiting that PDE isoform). The computational model does provide some evidence that the cAMP compartmentalization at TnI is important for maximal sarcomere shortening but the experimental evidence does not directly test this hypothesis.

As we note in the discussion (**p13, l21**), it is true that that maximal enhancement of contractility may be due to a mechanism other than reduced phosphorylation at TPNI, for example heterogeneity of cAMP signal at site(s) that are not monitored in this study. However, as this reviewer points out, the fact that the experimental findings are replicated by our simulations provides good supporting evidence for the model we propose. We have undertaken experiments as those suggested by this reviewer and our preliminary data suggest that multiple PDE isoforms are likely regulating the cAMP response at TPNI, making it more difficult to selectively manipulate the cAMP level locally using pharmacological tools. We will continue our efforts to address this point but, due to the complexity of this task, this will require significantly longer time than allowed for the revision of the current manuscript.

2. "The CUTie biosensor increases the spatial resolution cAMP sensing" is an overstatement and misleading. This biosensor improves the fidelity of the cAMP sensor when fused with other proteins but the resolution remains the same as previously developed probes. The introduction and discussion both make numerous references to resolution improvements but the only reference to "resolution" in the results comes in the first sentence. The improvement in fidelity though this type of biosensor design is interesting enough that overstating the spatial resolution detracts from the paper. This paper needs to be rewritten with more precise and accurate descriptions of the impact of the CUTie biosensor.

With 'increased spatial resolution' we intend to convey the concept that by targeting the sensor to specific multiprotein complexes we can monitor differences in cAMP between subcellular sites that are not otherwise distinguishable given the limit of optical resolution (about 200nm). The 'increased spatial resolution' is thus achieved because ideally all of the sensor resides at and all of the signal emanates from the target location. The CUTie on the longitudinal SR membrane surface creates a loose mesh around the myofilaments, where the CUTie on TPNI lies within tens of nanometres. Yet, we can readily discern the difference in local [cAMP] at these sites, below the optical resolution. For this reason we believe that the term 'spatial resolution' is appropriate in this case. On the other hand, the term 'fidelity' conveys the idea of the exactness, accuracy or specificity with which the signal is detected (e.g. with respect to detection of other molecules that may bind to the sensor). Although this is not an aspect that we specifically assess in this study, we don't expect the fidelity of CUTie for cAMP sensing to be improved with respect to other available reporters. Therefore in the revised manuscript we maintain the original description.

3. Computational model details are not included in the supplement of the paper. It is very important that the computational model details are published with the paper. While the models referenced have been previously published, this paper states that the model used involved merging two models, thus creating a unique model which must be fully explained. The validity of any model assumptions made when merging and modifying the previous models cannot be evaluated in this review as no details were provided.

In the Supplementary Data (**p4, l1**) we provided now more details on the model formulation describing the merging of the models and emphasizing that this new model recapitulates measured changes in AP, [Ca²⁺], and [Na⁺] in response to ISO.

Additional Points:

- *The authors hypothesize that cAMP compartmentalization in the TPNI is PDE driven. This hypothesis can be tested by synthetically compartmentalizing PDE with the CUTie biosensor, which they have already developed with their PDE4A1-CUTie construct. Thus, the Iso response of PDE4A1-CUTie should be compared to untargeted CUTie.*

As mentioned above (response to point 1, this reviewer) we are actively pursuing the identification of the PDEs involved in the regulation of cAMP levels at TPNI. However, it is unlikely that the approach suggested (use of PDE4A1-CUTie) will provide useful novel insight for a number of reasons. First, PDE4A1 may not be one of the PDE isoforms involved in the regulation of cAMP signals at TPNI. In addition, using fusions of CUTie with PDEs may give misleading results as it will be the presence of the PDE in close proximity to the sensor to dictate the level of cAMP detected, rather than the effect of any endogenous PDE or other local regulation. Other strategies (e.g. expression of catalytically inactive PDE isoforms) may on the contrary be more fruitful, once the nature of the relevant isoform(s) involved has been identified.

• *The decision to use 0.3 nM ISO was based on EPAC-SH187. What is the untargeted CUTie biosensor response to 0.3 nM ISO?*

The sensitivity of the CUTie reporter is not high enough to detect the small cAMP change generated by application of 0.3nM ISO (as we also show in Suppl Fig 6a for the targeted CUTies). The reason to choose such a low concentration of agonist was to avoid generation of an amount of cAMP in the cytosol that would exceed the amount generated by IBMX as in this circumstances one may expect a larger effect on stimulated contractility simply as a consequence of the larger overall amount of cAMP present in the cell.

• *What is meant by "global cAMP response" (pg 5 ln 32)?*

By 'global cAMP response' we mean the amount of cAMP that can be detected in the bulk cytosol by an untargeted, cytosolic sensor. We now clarify this in the text on **p4, l4** and **p8, l14**

• *Is it possible to validate the model predictions with the CUTie biosensor as well (i.e. MyBPC-CUTie)?*

This is another experiment that we are committed to perform and that will be part of a separate, follow-up study. Furthermore, as we discuss in the manuscript, while our results suggest that there may be differences in local cAMP even within the myofilaments (at TPNI vs. MyBPC), TPNI and MyBPC might be under the control of distinct kinase nano-domains (e.g. CaMKII, which also phosphorylates MyBPC at S282), or reduced local phosphatase activity may be responsible for the stronger phosphorylation of MyBPC.

Reviewers' comments:

Reviewer #1 (Remarks to the Author):

The authors have addressed all my comments satisfactorily. The manuscript is acceptable for publication.

Only one minor point. The disease model data are very nice and strengthen the paper significantly. Supplementary Table 1 should also include the units for all parameters measured, e.g. mm, % etc. HW/BW is usually expressed in mg/kg in the most literature.

Reviewer #2 (Remarks to the Author):

The authors have provided satisfactory answers to most of my comments. However, I am unsatisfied, and I must say a little bit annoyed, with the reply the authors made to my first comment. They claim that they were unable to perform the calibration experiments in ARVMs because the cells hypercontracted and died shortly after establishing the whole cell configuration. However, intracellular dialysis of ARVMs have been successfully achieved by a number of other groups and I see no reason why the authors failed in their attempts. While I imagine a number of technical reasons why they failed in performing these important experiments (insufficient gigaseal, inadequate composition of the pipette solution, abnormal fragility of the cells...), all of them could be circumvented if the authors had tried harder. Without the demonstration that the concentration-response curves for the FRET changes in response to cAMP are identical for the three CUTie probes when expressed in ARVMs, the differences seen between the signals obtained in the TPNI compartment and the LTCC and PLB compartments cannot be interpreted.

Reviewer #3 (Remarks to the Author):

The authors did a good job addressing concerns about the previous disconnections between the concentrations of ISO used for different assays. However, there are still two major issues about how the research is presented with respect to the improvement on spatial resolution and the description of the computational model.

Major Concerns:

1. The spatial resolution of the targeted CUTie sensors is not increased compared to any other targeted cAMP sensor using the same targeting motif. The novel aspect of the targeted CUTie sensors is that the dynamic range is unaffected by the incorporation of the targeting motif thus enabling a more direct and accurate comparison between differentially targeted reporters. While this improved fidelity of the CUTie sensor does increase the confidence in the measured responses from different compartments, the ability to directly resolve distinct cAMP concentrations in space is not different than any other cAMP sensor using the same targeting motif. While the "back-of-the-envelope" estimation of the possible size of a cAMP microdomain based on the textbook values of cardiomyocyte dimensions and differences in CUTie responses is an interesting discussion point, this does not support the claim that the CUTie based method achieves unprecedented spatial resolution. Thus, the relevant sections should be re-written to avoid over-claims.

2. The inclusion of Supp. Fig 7 does clarify the model comparisons discussed in the paper and the description of the mathematical model development is improved from the original submission of the paper but there are still insufficient details of the model included in the supplement. It would be impossible for an independent researcher to recreate the model and replicate the results with the information provided in the paper. The details of the computational model should be at the very least to the same level as that provided in Negroni, J.A. et al JMCC 2015 (e.g. parameter values, changed equations).

Minor points:

1. Line 179 – “ the untargeted cAMP sensor” – it would be more clear if EPAC-SH187 were used here instead to be more clear as to what sensor is used.
2. Line 276 – acute catecholamine stimulation is generally not considered “stress”. This term is better reserved for the discussion on the chronic stimulation or MI experiments.

Point-by-point response to the referees' comments:**Reviewer #1:**

The disease model data are very nice and strengthen the paper significantly. Supplementary Table 1 should also include the units for all parameters measured, e.g. mm, % etc. HW/BW is usually expressed in mg/kg in the most literature.

Units have now been included

Reviewer #2:

The authors have provided satisfactory answers to most of my comments. However, I am unsatisfied, and I must say a little bit annoyed, with the reply the authors made to my first comment. They claim that they were unable to perform the calibration experiments in ARVMs because the cells hypercontracted and died shortly after establishing the whole cell configuration. However, intracellular dialysis of ARVMs have been successfully achieved by a number of other groups and I see no reason why the authors failed in their attempts. While I imagine a number of technical reasons why they failed in performing these important experiments (insufficient giga-seal, inadequate composition of the pipette solution, abnormal fragility of the cells...), all of them could be circumvented if the authors had tried harder. Without the demonstration that the concentration-response curves for the FRET changes in response to cAMP are identical for the three CUTie probes when expressed in ARVMs, the differences seen between the signals obtained in the TPNI compartment and the LTCC and PLB compartments cannot be interpreted.

We agree with reviewer #2 that repeating the full calibration of the CUTie sensors in ARVM (as was shown for CHO cells in Fig 2f) would be ideal. However, in NRVM (Suppl Fig 2c) and ARVM (new Suppl Fig 2d-e) we could only obtain partial calibrations, largely because of very high constitutive adenylyl cyclase and PDE activity in myocytes. To control intracellular [cAMP] during cell-attached patch-clamp dialysis for myocyte calibrations required PDE inhibition by IBMX (Suppl Figs 2c-e). The Panel at right shows that PDE inhibition *per se*, increased baseline FRET to about 35% of the maximal FRET change, and this was similar for all three targeted sensors (first 3 bars in Suppl Fig 2c). This was also comparable to the 10 μ M cAMP signal (for all three CUTies; next 3 bars) and made calibration impossible below 10 μ M cAMP. However, from 10 μ M cAMP up to saturation (1 mM cAMP and with forskolin) FRET for all 3 sensors behaved identically in NRVMs (Suppl Fig 2c).

This same approach in ARVMs was much more difficult because (as previously noted), ARVMs became unstable when dialysed with [cAMP] > 10 μ M with IBMX, which is worse for higher [cAMP] in the patch pipette, usually resulting in myocyte death. We do not understand why this happens, but spent several weeks trying to overcome this issue (see end of this response).

The only conditions where we obtained stable useful results was at reduced bath [Ca²⁺] (to 1mM) with 30 μ M cAMP + IBMX in the patch pipette (higher [cAMP] caused instability). These new experiments are now in **Suppl Fig 2e** and show no significantly difference among signals for the 3 targeted sensors for this sub-maximal activation.

To further assess graded [cAMP] responses over a broad [cAMP] range where cytosolic gradients are not expected in intact ARVMs, we used graded adenylyl cyclase activation (50 nM to 25 μ M forskolin) with PDE blocked by IBMX (new **Suppl Fig 2d**; also at right). 50 nM forskolin failed to raise [cAMP] above the IBMX baseline (for any of the 3 CUTies), but increasing [forskolin] produced very similar FRET signals for all 3 sensors in intact ARVMs. While

we would have preferred to include absolute [cAMP] CUTie calibrations in adult ventricular myocytes, these results show how consistently these novel targeted sensors behave, independent of the targeting protein.

We did make several attempts to mitigate hypercontraction and cell death by changing the composition of the intracellular/extracellular buffers and by including cytochalasin D in the pipette solution. We suspect there is some consequence of uncontrolled PKA activation by combining PDE inhibition and dialysis with high global [cAMP] that causes instability and cell death. We now think this cell instability and myocyte death was not due to known technical reasons (e.g. insufficient giga-seal, inadequate composition of the pipette solution, abnormal fragility of the cells etc, as suggested by this reviewer). We carefully monitored seal resistance, which was $6.2 \pm 0.5 \text{ G}\Omega$ (mean \pm SEM, $n = 42$), and membrane potential values were reasonable (near -65 mV or more negative). In addition, the **representative trace at right** shows that the myocyte is OK even when dialyzed with $100 \mu\text{M}$ cAMP (although FRET increased only slightly, due to the activity of PDEs). However, when IBMX was applied to inhibit PDEs, the cell hypercontracted and died. This effect may merit further analysis, but is beyond the practical scope of this study.

Reviewer #3:

The authors did a good job addressing concerns about the previous disconnections between the concentrations of ISO used for different assays. However, there are still two major issues about how the research is presented with respect to the improvement on spatial resolution and the description of the computational model.

Major Concerns:

1. The spatial resolution of the targeted CUTie sensors is not increased compared to any other targeted cAMP sensor using the same targeting motif. The novel aspect of the targeted CUTie sensors is that the dynamic range is unaffected by the incorporation of the targeting motif thus enabling a more direct and accurate comparison between differentially targeted reporters. While this improved fidelity of the CUTie sensor does increase the confidence in the measured responses from different compartments, the ability to directly resolve distinct cAMP concentrations in space is not different than any other cAMP sensor using the same targeting motif. While the “back-of-the-envelope” estimation of the possible size of a cAMP microdomain based on the textbook values of cardiomyocyte dimensions and differences in CUTie responses is an interesting discussion point, this does not support the claim that the CUTie based method achieves unprecedented spatial resolution. Thus, the relevant sections should be re-written to avoid over-claims.

In the revised text we have removed the wording ‘spatial resolution’ and substituted with ‘accuracy’ (abstract, line 7), ‘accuracy and fidelity’ (p. 4, line 21 and p. 12, line 3).

2. The inclusion of Supp. Fig 7 does clarify the model comparisons discussed in the paper and the description of the mathematical model development is improved from the original submission of the paper but there are still insufficient details of the model included in the supplement. It would be impossible for an independent researcher to recreate the model and replicate the results with the information provided in the paper. The details of the computational model should be at the very least to the same level as that provided in Negroni, J.A. et al JMCC 2015 (e.g. parameter values, changed equations).

We agree with the reviewer that it is important to provide sufficient information about the model that an independent researcher could reproduce our results. For this reason, we provide (as in all of our papers) a link to an online model repository with our code (uploads are made upon publication,

but we are attaching it here for review purposes). In our experience, this is the most reliable and efficient way to replicate simulations from other papers, without incurring typographical errors, missing parameters or equations, etc. We have also revised the Modeling section of the on-line Supplement to further clarify this point and several other ones that should make the model more understandable for the reader. We emphasize more clearly that:

- The ionic and Ca handling model is unaltered from the original Morotti *et al.* formulation (*J Physiol* 2014)
- The contractile model is unchanged from the original Negroni *et al.* formulation (*J Mol Cell Cardiol* 2015)
- We specified the changes in the mouse model when including PKA-dependent phosphorylation of the myofilament proteins (TPNI, MyBPC, and titin). To do so, we adapted the equations in the Negroni *et al.* model to reproduce PKA phosphorylation extent and kinetics used for TPNI in the Morotti *et al.* model.

We specified the parameter changes in the above equations in the various conditions simulated (IBMX, and ISO with reduced cAMP at all or each myofilament targets).

Minor points:

1. Line 179 – “the untargeted cAMP sensor” – it would be more clear if EPAC-SH187 were used here instead to me more clear as to what sensor is used.

Amended as suggested

2. Line 276 – acute catecholamine stimulation is generally not considered “stress”. This term is better reserved for the discussion on the chronic stimulation or MI experiments.

The word ‘stress’ has been substituted with ‘catecholamines’ (p. 12, line 12)